# ON THE CAPACITY OF SELF-ATTENTION

## ABSTRACT

While self-attention is known to learn relations among tokens, we lack a formal understanding of its capacity: how many distinct relations can a single layer reliably recover for a given budget?

To formalize this, we introduce *Relational Graph Recognition* (RGR), where the key-query channel represents a graph on $m$ items with $m'$ directed edges, and, given a context of items, must recover the neighbors of each item. We measure resources by the *total key dimension* $D_K = h\,d_k$. Within this framework, we analytically derive a capacity scaling law and validate it empirically. We show that $D_K = \Theta(m' \log m'/d_{\text{model}})$ is both necessary (information-theoretic lower bound) and sufficient (explicit construction) in a broad class of graphs to recover $m'$ relations. This scaling law directly leads to a new, capacity-based rationale for multi-head attention that applies even when each item only attends to a single target. When embeddings are uncompressed ($m = d_{\text{model}}$) and the graph is a permutation, a single head suffices. However, compression ($m > d_{\text{model}}$) forces relations into overlapping subspaces, creating interference that a single large head cannot disentangle. Our analysis shows that allocating a fixed $D_K$ across many small heads mitigates this interference, increasing the number of recoverable relations. Controlled single-layer experiments mirror the theory, revealing a sharp performance threshold that matches the predicted capacity scaling and confirms the benefit of distributing $D_K$ across multiple heads.

Altogether, these results provide a concrete scaling law for self-attention capacity and a principled design rule for allocating key-query budget across heads.

## 1 INTRODUCTION

The transformer architecture, and its self-attention mechanism in particular, has revolutionized fields from natural language processing to computer vision (57). At its core, self-attention computes a similarity-weighted pattern of pairwise relationships among items in a context: queries match keys; the resulting scores route information via values (50; 35; 69; 58; 15; 54). We ask a basic question: for a fixed self-attention mechanism size, how many target relationships can a single attention layer represent and reliably recover? We call this the layer's *capacity*. Capacity is a foundational property for several reasons. (i) It imposes a hard ceiling on relational computation: beyond a threshold, no training procedure or dataset can make a layer uniformly recover all relations, much like rank bounds in linear models. (ii) It complements mechanistic work that isolates specific attention circuits in trained transformers (9; 60; 44; 32), by asking how many independent circuits can coexist. (iii) It provides an actionable resource scaling law, by describing how relationship capacity grows with increasing attention mechanism budget.

We also demonstrate that capacity impacts when increasing the number of heads is useful. Multiple heads are often conceived as a way for a source concept to attend to multiple different targets (57), but looking at self-attention through the lens of capacity demonstrates that multiple heads are beneficial even in the simple case where each concept attends to only a single target. Specifically, when compressed embeddings

are used, many relations must be stored in overlapping subspaces; distributing the self-attention budget across many small heads reduces interference and increases the number of relations that can be cleanly separated—consistent with both pruning studies and expressivity results for attention (61; 43; 10).

One might hope to answer capacity empirically by probing large trained models. In practice, this is ill-posed. Modern transformers superpose many relationships in shared subspaces; heads are polyfunctional and context-dependent, so the number of "active" relations is not directly observable. Moreover, attention weights need not align with causal importance (27), and even sophisticated circuit-tracing pipelines currently miss parts of the QK computation that determine *where* a head attends (32). Beyond these methodological issues, superposition makes enumeration intrinsically hard: models can store more features than basis directions, packing multiple concepts into overlapping subspaces (17; 7). As a result, interpretability work thus far has not revealed how many relationships can be supported by a fixed attention budget.

We therefore introduce a *framework*—**Relational Graph Recognition (RGR)**—and analyze an idealized self-attention *model* for solving RGR. The framework allows us to explicitly control both the structure and the number of attention relationships by casting self-attention as recovering edges of a relational graph among $m$ items, while the model preserves the computational constraints and symmetries of attention. This allows predictions through principled analysis as well as controlled simulations that directly test those predictions. Our abstraction isolates the *key–query* computation that determines *where* a head attends, separating it from the OV pathway that determines *what* is written—a split made explicit in recent mechanistic analyses of attention heads (32). As a result, our attention budget is defined in terms of the *total key dimension* $D_K = h\,d_k$, where $h$ is the number of heads and $d_k$ the per-head key (and query) width.

**Problem Formulation: Relational Graph Recognition (RGR)**    To make "relationships" precise, we cast the core task of self-attention as a graph recovery problem.

**Task.**    Let $G = (V, E)$ be a directed graph on $m = |V|$ items with $m' = |E|$ edges. A *context* is an ordered tuple $\mathcal{C} = (v_{i_1}, \ldots, v_{i_\ell})$ with $1 \le \ell \le m$. The **Relational Graph Recognition (RGR)** mapping takes $(G, \mathcal{C})$ and, for each $v \in \mathcal{C}$, returns its in-context neighbors $N_G(v; \mathcal{C}) = \{v' \in \mathcal{C} : (v, v') \in E\}$. Given a graph $G$, we wish to find a parameterization $\Theta(G)$ such that the mapping defined by $\Theta(G)$ correctly produces $N_G(v; \mathcal{C})$ for *every* context $\mathcal{C}$ and every $v \in \mathcal{C}$.

**Capacity question.**    Fix an input embedding dimension $d_{\mathrm{model}}$. For a graph family $\mathcal{G}_{m,m'} = \{\, G : |V| = m,\ |E| = m' \,\}$, we ask for the minimal *total key dimension* $D_K = h\,d_k$ (number of heads times per-head key/query width) such that the self-attention model in Section 3 can realize the RGR mapping for all $G \in \mathcal{G}_{m,m'}$ and all contexts $\mathcal{C}$. We refer to this minimal $D_K$ as the *capacity required* by $\mathcal{G}_{m,m'}$ at embedding dimension $d_{\mathrm{model}}$.

**Why this abstraction.**    RGR isolates the key–query channel that determines *where* attention goes, while preserving the permutation symmetries and parameter sharing of self-attention[1]. It lets us dial graph complexity $(m, m')$ and the budget $D_K$ independently, enabling the information-theoretic bounds, constructive designs, and targeted experiments reported below. Exact mechanics (score computation, aggregation across heads and context are specified in Section 3. Our core model omits both softmax and values, but we also demonstrate that those can be added to the model without altering our main conclusions.

---

[1]We also describe below how positional embeddings can be incorporated into the model.

SUMMARY OF RESULTS

Our analysis yields both fundamental limits and constructive proofs of capability for self-attention as a relational reasoner, and our experiments validate the predictions in an idealized setting that mirrors the theoretical model. The main contributions are:

**Formal model and budget.** We cast "where to attend" as *Relational Graph Recognition (RGR)* and analyze an idealized attention layer that preserves attention symmetries while isolating the key–query computation. The complexity measure is the total key dimension $D_K = h\, d_k$ (Sec. 3).

**Information-theoretic lower bound.** We show that recovering $m'$-edge graphs on $m$ items with fixed margin requires $D_K = \Omega\!\left(\frac{m'}{d_{\text{model}}} \log \frac{m^2}{m'}\right)$, independent of parameter precision and for any context length $\ell \geq 2$ (Sec. D of the Appendix). This formalizes the intuition that the key-query matrices need to express sufficient information to describe the underlying relational graph. It also shows that representing more relationships demands greater key–query capacity, and that a smaller embedding dimension requires a larger total key dimension.

**Asymptotically Optimal Constructions.** We provide explicit attention-based algorithms for RGR within the idealized model (Section 4). We achieve $D_K = O\!\left(\frac{(m'+\Delta)}{d_{\text{model}}} \log m'\right)$, where $\Delta$ is the maximum degree of the graph $G$. Under a mild condition around balanced degrees $((\Delta/d_{\text{avg}} \leq m/d_{\text{model}})$, this closes the gap between upper and lower bounds for all but very dense graphs, which have a gap factor of $\log m$. These constructions assume random Gaussian unit norm embeddings and also extend to any embedding satisfying a superposition hypothesis style near-orthogonality condition. These constructions also surface the core computational principles of self-attention and serve as concrete, testable hypotheses about the internal mechanisms transformers might learn. These constructions are presented in a simplified model of attention, without softmax or a value channel, but we demonstrate in Appendices A and B that these factors can be added to the model with limited impact to these results.

**Capacity-Based Rationale for Multi-Head Attention.** Our analysis shows that multiple heads are beneficial even when each source has a single correct target (Section 5): when using compressed embeddings $d_{\text{model}} \ll m$, the signals for different relationships are superposed and multiple heads mitigate interference by specializing on disjoint subsets of relationships, thereby allowing the per-head dimension to be small. This provides a principled capacity-centric justification for multi-head attention as a method to reduce noise. We also demonstrate, both theoretically and empirically, that the multi-head advantage is surprisingly robust to model details: it persists when we add softmax (with or without scaling), include a value channel, and vary graph density.

**Empirical Validation of Capacity and Head Scaling.** In controlled single-layer experiments that mirror the idealized model, performance exhibits a *sharp* transition as $D_K$ increases (Section 6). The minimal $D_K$ needed to reach high accuracy (micro-F1 $\gtrsim 0.99$) grows rapidly with $m$ and shrinks with $d_{\text{model}}$, consistent with the theory. We recover the predicted scaling: $D_K^\star \approx C\,\frac{m \log m}{d_{\text{model}}}$ with a single global threshold at test time. A one-parameter fit to all data yields $C = 1.19$ with $R^2 = 0.944$. Discarding three data points that the theory predicts to perform slightly worse yields $C = 0.966$ with $R^2 = 0.992$. We also find a pronounced *multi-head advantage* even for permutation graphs: the smallest passing models use several heads while keeping per-head width small and the optimal head count scales linearly with $\frac{m}{d_{\text{model}}}$, as predicted by the theory. We also see that capacity is largely insensitive to context length over $\ell \in \{16, 32\}$. Finally, we test the impact of increasing graph density with $r$-regular graphs. By varying $r$, we see that, consistent with the theory, it is the number of edges, not vertices, that dictates both the scaling of $D_K$ and the optimal head count.

Together, these findings yield a quantitative scaling law for capacity as a function of $D_K$ and $d_{\text{model}}$, and reveal a principled *multi-head advantage* even when each source has only a single correct target—clarifying when and how to allocate key–query budget across heads. The close alignment between our constructive bounds and empirical thresholds provides a concrete, falsifiable foundation for the computational principles that enable self-attention.

## 2 RELATED WORK

Given the breadth of prior work on attention, we defer an extended survey to Appendix H, covering expressivity, language-theoretic limits, connectivity, memorization, superposition, interpretability, and graph-structured models.

Closest to our focus are works on *memorization capacity* (55; 34; 31), including analyses of memorization in attention modules (41). While aligned in spirit, the problem formulations are different: memorization typically maps each *context* to a single output token/label, whereas our RGR setting asks for the recovery of in-context neighbors for *every* context from a set of possible tokens. Reductions between the two would require memorization handling a combinatorial number of contexts (polynomial in $m$ for fixed $\ell$, and exponential when $\ell$ scales with $m$), and we are not aware of efficient reductions that preserve guarantees in either direction. Accordingly, bounds in one setting do not directly imply bounds in the other. Not surprisingly, capacity results for memorization provide different scaling laws then ours. For example they often include explicit dependence on sequence length $\ell$ (41; 34; 31), whereas in our analysis and experiments, length plays a limited role. Our abstraction isolates the key–query *addressing* step—"where to attend"—which mechanistic analyses identify as central to head routing (32). In this sense, RGR complements parameter-centric memorization settings that emphasize "what to output": we target the capacity required to *select* the correct neighbors across contexts.

Beyond memorization, prior theory characterizes *what* Transformers can compute with sufficient resources—universality/approximation (67), fine-grained attention-matrix expressivity (38), and structural bottlenecks such as per-head low rank and rank collapse without mixing (5; 14). Language-theoretic and composition results map limitations at fixed budgets (23; 45); orthogonally, restricting connectivity rather than dimensions shows universality of $O(\ell)$-sparse patterns and principled pruning of dense ones (68; 64); and algorithmic views analyze in-context procedures (37). Mechanistic studies find head specialization and prune-ability (9; 43), while memory-centric views link attention and FFNs to associative/key–value memories (47; 21). See Appendix H for more details.

## 3 MODELING THE SELF-ATTENTION MECHANISM

We model the *key–query (QK)* channel of a single self-attention layer for RGR, retaining permutation symmetry and parameter sharing while omitting components that do not affect the binary edge decision (softmax, $1/\sqrt{d_k}$ scaling, and values; see Appendix. G). The input is an ordered context of distinct vertices of length $\ell \leq m$, given as $\mathcal{C} = (v_{i_1}, \ldots, v_{i_\ell})$, one each to $\ell$ attention units. Each $v \in V$ is described using a unique embedding $\mathbf{x}_v \in \mathbb{R}^{d_{\text{model}}}$. Positional information is not explicitly modeled; if needed, positions can be incorporated by treating $(\text{token}, \text{position})$ as distinct vertices.

**Single head.** Each attention unit with a single head uses the same shared projection matrices $W_Q, W_K \in \mathbb{R}^{d_{\text{model}} \times d_k}$. For each $v_{i_p} \in \mathcal{C}$, $\mathbf{q}_{i_p} = \mathbf{x}_{i_p} W_Q$, $\mathbf{k}_{i_p} = \mathbf{x}_{i_p} W_K$. The unnormalized score from source $v_{i_p}$ to target $v_{i_q}$ is $S_{pq} = \mathbf{q}_{i_p} \cdot \mathbf{k}_{i_q}^{\top}$. We declare an edge $(v_{i_p}, v_{i_q})$ present iff $S_{pq} > \tau$ for a global threshold $\tau$. Only pairs inside $\mathcal{C}$ are tested.

**Multi-head extension.** With $h$ heads, each head $k$ has $(W_Q^{(k)}, W_K^{(k)}) \in \mathbb{R}^{d_\text{model} \times d_k}$ and produces $S_{pq}^{(k)}$. In our core model, we aggregate per pair by $S_{pq}^\text{max} = \max_{k \in \{1,\dots,h\}} S_{pq}^{(k)}$, and decide $(v_{i_p}, v_{i_q}) \in E \Leftrightarrow S_{pq}^\text{max} > \tau$. This "OR-of-heads" view matches the common specialization picture. Also, replacing $\max$ by log-sum-exp yields an equivalent classifier after a global threshold shift; see App. G. We also demonstrate in Appendix A how to incorporate softmax (with or without scaling). In Appendix B we present a model and results that incorporate a value channel as well.

**Algorithmic objective and budget.** A *construction for RGR* maps a graph $G = (V, E)$ to weights $\{(W_Q^{(k)}, W_K^{(k)})\}_{k=1}^h$ and a threshold $\tau$ that realize the correct edge decisions for *all* contexts $\mathcal{C}$, regardless of length $\ell$. We measure complexity by the *total key dimension* $D_K = h\, d_k$, since QK is implemented as two batched multiplications with concatenated weights $W_Q^\text{cat} = [W_Q^{(1)} | \cdots | W_Q^{(h)}]$ and $W_K^\text{cat} = [W_K^{(1)} | \cdots | W_K^{(h)}]$ in $\mathbb{R}^{d_\text{model} \times (h d_k)}$; the dominant cost and parameter footprint scale with $h\, d_k$ rather than $h$ or $d_k$ alone (see Appendix G).[2] Our goal is to minimize $D_K$ over a graph family $\mathcal{G}_{m,m'}$ for a given $d_\text{model}$.

In Appendix G, we provide further justification for many of our model choices. Specifically, we address the use of thresholding instead of scaling/softmax, our head aggregation rule, and the lack of a value pathway.

## 4 EXPLICIT CONSTRUCTIONS FOR RELATIONAL GRAPH RECOGNITION

In this section, we provide explicit constructions for the attention weights that solve RGR within the idealized model of Section 3. These constructions establish an upper bound on the $D_K$ required to solve RGR, providing a concrete measure of the self-attention mechanism's capacity for this task. Each design is proven to perform with high probability (w.h.p.), meaning at least $1 - m^{-\gamma}$ for some constant $\gamma > 2$. We begin with a brief warm-up sketch—recognizing permutation graphs with one-hot embeddings—then move to compressive embeddings. In Appendix F, we show how to generalize these result to arbitrary graphs and general embeddings. Throughout, $i$ indexes the *source* and $j$ a *candidate target* in $E$; keys are tied to targets and queries are tied to sources.

**Construction I: Permutation Graphs with One-Hot Embeddings (Warm-up/sketch).** We consider a permutation graph $G$ on $m$ items with edges $E = \{(v_i, v_{\pi(i)})\}$ and one-hot node encodings $\mathbf{x}_i = \mathbf{e}_i \in \mathbb{R}^m$ (so $d_\text{model} = m$). A single head ($h = 1$) suffices. Assign each target $v_j$ a random binary signature as its key $\mathbf{k}_j$ by drawing $W_K \in \{0,1\}^{m \times d_k}$ with i.i.d. Bernoulli($p$) entries (e.g., $p = 1/4$) and setting $\mathbf{k}_j = \mathbf{e}_j W_K$. For each source $v_i$, set the query to the signature of its target: $\mathbf{q}_i = \mathbf{k}_{\pi(i)}$, i.e., the $i$-th row of $W_Q$ equals the $\pi(i)$-th row of $W_K$. With $d_k = C \log m$ and threshold $\tau = \frac{p+p^2}{2} d_k$, one has

$$S_{i,\pi(i)} = \mathbf{k}_{\pi(i)} \cdot \mathbf{k}_{\pi(i)} \sim \text{Binomial}(d_k, p), \qquad S_{ij} = \mathbf{k}_{\pi(i)} \cdot \mathbf{k}_j \sim \text{Binomial}(d_k, p^2) \ (j \neq \pi(i)).$$

Chernoff bounds and a union bound over all $(i,j)$ yield simultaneous separation $S_{i,\pi(i)} > \tau > S_{ij}$ w.h.p. when $d_k = \Theta(\log m)$, so a single head recovers all edges. This matches the $\Omega(\log m)$ lower bound (Appendix D) and the same argument extends to softmax scoring (Appendix G). Full details—algorithm and proof—are deferred to Appendix F as a "warm-up" proof; Construction II below generalizes this scheme to compressive embeddings, but first we point out that a separation of the the type described is sufficient to handle contexts of any length.

**Lemma 4.1** (Context-robustness). *Fix parameters $\{(W_Q^{(k)}, W_K^{(k)})\}_{k=1}^h$ and a threshold $\tau$. Let the aggregated score be $S_{ij}^\text{max} := \max_k S_{ij}^{(k)}$. If, simultaneously for all $i \in V$,*

$$S_{i,\pi(i)}^\text{max} > \tau \quad and \quad S_{ij}^\text{max} < \tau \text{ for all } j \neq \pi(i), \tag{$\star$}$$

---

[2] All statements remain (up to a factor of two) if one counts $D_Q + D_K$.

*then for every context $\mathcal{C} \subseteq V$ and every source $i \in \mathcal{C}$: (i) if $\pi(i) \in \mathcal{C}$ then $S_{i,\pi(i)}^{\max} > \tau$ and $S_{ij}^{\max} < \tau$ for all $j \in \mathcal{C} \setminus \{\pi(i)\}$; (ii) if $\pi(i) \notin \mathcal{C}$ then $S_{ij}^{\max} < \tau$ for all $j \in \mathcal{C}$. Hence the same parameters recognize $E|_{\mathcal{C}}$ for every $\mathcal{C}$ and every length $\ell$.*

*Proof.* Restricting from $V$ to $\mathcal{C}$ only removes candidate targets. If $\pi(i) \in \mathcal{C}$, the inequalities in $(\star)$ remain true after removing all $j \notin \mathcal{C}$. If $\pi(i) \notin \mathcal{C}$, every remaining $j \in \mathcal{C}$ satisfies $j \neq \pi(i)$, so $S_{ij}^{\max} < \tau$ by $(\star)$. $\qquad\square$

**Construction II: Permutations Under Compressive Embeddings**  We now extend the permutation case to the compressive regime $d_{\text{model}} \ll m$ under a *Gaussian unit-norm* embedding.[3] Each item $v_i$ is embedded as a fixed vector $\mathbf{x}_i \in \mathbb{R}^{d_{\text{model}}}$ drawn i.i.d. as $\tilde{\mathbf{x}}_i \sim \mathcal{N}(0, I/d_{\text{model}})$ and then $L_2$-normalized, i.e., $\mathbf{x}_i = \tilde{\mathbf{x}}_i / \|\tilde{\mathbf{x}}_i\|_2$. Write $X \in \mathbb{R}^{m \times d_{\text{model}}}$ for the matrix with $i$-th row $\mathbf{x}_i^\top$. Given such an embedding and permutation $\pi$, our goal is to construct attention parameters that recognize $G$.

**Multi-Head Algorithmic Construction**  The fundamental challenge with embeddings is that the input $\mathbf{x}_i$ is a superposed representation of the node's identity. Our construction first approximately inverts the embedding process, projecting the $d_{\text{model}}$-dimensional vector $\mathbf{x}_i$ back into the $m$-dimensional one-hot space using the transpose of the embedding matrix. We then apply the logic from the one-hot case. However, doing this with a single head yields too much noise due to the inversion being only approximate. We mitigate this noise by using multiple attention heads, where each is responsible for recognizing the outgoing edges from a disjoint subset of sources. This results in smaller individual heads, and thus less noise. The proof of the following theorem appears in Appendix F, where we also show how to extend these results to more general embeddings and graphs.

---

**Algorithm 1** Construction for Permutation Graphs with Compressive Embeddings

---

1: **Input:** Permutation graph $G = (V, E)$ with $\pi : V \to V$; embedding matrix $X \in \mathbb{R}^{m \times d_{\text{model}}}$.
2: **Parameters:** Number of heads $h = \frac{m}{d_{\text{model}}}$; per-head key/query dimension $d_k = C \log m$ for a sufficiently large absolute constant $C$. **Set Threshold:** $\tau = \frac{1}{2} d_k$.
3: **Partition sources and targets:** Split $V$ into $h$ disjoint blocks $V_1, \ldots, V_h$ of size $|V_k| = d_{\text{model}}$. For each head $k$, define its target set $T_k := \{\pi(s) : s \in V_k\}$, which is a permutation of $V_k$. Head $k$ is responsible for sources in $V_k$ and targets in $T_k$ (a permutation of $V_k$).
4: **Random signatures:** Draw $W_{\text{sig}} \in \{\pm 1\}^{m \times d_k}$ with i.i.d. Rademacher entries; let $\mathbf{w}_j$ be its $j$-th row.
5: **Ideal one-hot-space templates (for each head $k$):**
6:     **Query Matrix:** $W'_{Q,(k)} \in \mathbb{R}^{m \times d_k}$ with row $i$ equal to $\mathbf{w}_{\pi(i)}$ if $i \in V_k$, and $\mathbf{0}$ otherwise.
7:     **Key Matrix:** $W'_{K,(k)} \in \mathbb{R}^{m \times d_k}$ with row $j$ equal to $\mathbf{w}_j$ if $j \in T_k$, and $\mathbf{0}$ otherwise.
8: **Project back to model space (approximate de-embedding):**

$$W_Q^{(k)} = X^\top W'_{Q,(k)}, \qquad W_K^{(k)} = X^\top W'_{K,(k)}.$$

---

**Theorem 4.2** (Multi-head recognition under Gaussian unit-norm embeddings)**.** *Assume the setup above with $h = \frac{m}{d_{\text{model}}}$ heads, per-head dimension $d_k = C \log m$ for a sufficiently large absolute constant $C$, and*

---

[3]Random spherical codes are nearly orthogonal—inner products concentrate tightly around 0—which lets us obtain clean dot-product thresholds and $O(\log m)$ scaling in our separation arguments. This cosine-geometry is also standard and effective in practice: many systems explicitly constrain features to a hypersphere (e.g., NormFace and ArcFace in face recognition; Spherical Text Embedding in NLP; spherical objectives in metric learning). See (59; 12) for concentration/JL background and (63; 13; 42; 70) for representative uses of unit-sphere embeddings.

*threshold $\tau = \frac{1}{2}d_k$. If $d_{\text{model}} \geq c_0 \log m$ for a sufficiently large absolute constant $c_0$, then with probability at least $1 - m^{-3}$ over the draw of $(X, \text{signatures})$,*

$$\forall i \in V \; \exists k \in [h] \text{ with } i \in V_k: \quad S^{(k)}_{i,\pi(i)} > \tau \quad \text{and} \quad S^{(k)}_{ij} < \tau \;\; \forall j \neq \pi(i). \tag{1}$$

*Consequently, max-pooling over heads correctly recognizes all edges and $D_K = h\,d_k = \Theta\!\left(\frac{m \log m}{d_{\text{model}}}\right)$.*

**Consequences.** The total key budget satisfies $D_K = O\!\left(\frac{m \log m}{d_{\text{model}}}\right)$, which matches our lower bound up to constants. Moreover, the non-edge bounds hold head-wise, so for any $j \neq \pi(i)$ we have $S^{(k)}_{ij} < \tau$ for all $k$ and hence $S^{\max}_{ij} < \tau$, while $S^{\max}_{i,\pi(i)} > \tau$ at the true target. By Lemma 4.1, the same parameters correctly recognize $E|_{\mathcal{C}}$ for every context $\mathcal{C}$ and for every context length. (As in Construction I, the thresholding analysis translates to softmax; see Section G.)

## 5   THE POWER OF MULTIPLE HEADS

With no compression (Construction I), a single head suffices: queries and keys can coincide exactly on true edges and be nearly orthogonal otherwise, yielding true-edge scores $\Theta(d_k)$ and non-edge scores concentrated near 0. In the compressive setting (Construction II), we first approximately de-embed $\mathbf{u}_i := \mathbf{x}_i X^\top = \mathbf{e}_i + \boldsymbol{\delta}_i$, so each source carries a small *leakage* vector $\boldsymbol{\delta}_i$ that spreads mass across many coordinates. With Rademacher signatures (see §4) the head-$k$ score decomposes into a signal term—$\Theta(d_k)$ for true edges and concentrated near 0 for non-edges—and a noise term controlled by the leakage. The dominant component of this noise, denoted $N_3$ in Appendix F, scales with the *block size* $B := |T_k|$ served by a head. Intuitively, if the block size is too large, there is too much noise, and so multiple heads are required to keep the block size small.

$$N_3(B) \;\asymp\; \frac{B}{d_{\text{model}}}\,\sqrt{d_k \log m}. \tag{2}$$

To guarantee (w.h.p.) a fixed margin between the true target and all non-targets, it must be that, for constants $c_1, c_2 > 0$,

$$N_3(B) \;\leq\; c_1 d_k \quad \Longrightarrow \quad d_k \;\geq\; c_2\,\frac{B^2}{d_{\text{model}}^2}\,\log m. \tag{3}$$

**Single head with compression.** If one head serves all items, then $B = m$. Applying (3) gives the requirement $d_k \geq c_2\,\frac{m^2}{d_{\text{model}}^2}\log m$, and since $h = 1$ here, the total key dimension is $D_K = d_k$.

**Multiple heads with compression.** Construction II partitions the items into $\frac{m}{d_{\text{model}}}$ heads with $B = d_{\text{model}}$ per head. Plugging $B = d_{\text{model}}$ into (2) yields $N_3(B) = \Theta(\sqrt{d_k \log m})$. Taking $d_k = c_3 \log m$ with $c_3$ larger than the constant in (2) ensures $N_3 \leq c_1 d_k$ w.h.p., and the total key dimension is $D_K = h\,d_k = O\!\left(\frac{m}{d_{\text{model}}}\log m\right)$.

**Consequence.** As a result, in the compressive regime, if $m = \omega(d_{\text{model}})$, the single head requirement above implies asymptotically larger $D_k$ than the multihead construction. Or, equivalently, for a fixed $D_k$ budget, multiple heads can handle more edges (relationships) than a single head, even in a permutation graph. Multiple heads do not boost per-head expressivity; they *localize* de-embedding noise by reducing block size $B$, so that each head aggregates leakage over only fewer coordinates, bringing the noise to a manageable level. Note that this is not a lower bound for *all* conceivable single-head designs, but it shows that within the de-embedding to signature template we use, a single head cannot perform as well as multiple heads.

## 6 EXPERIMENTS

We conduct experiments in an *idealized self-attention* setting, mirroring our theoretical model, to test several predictions. First, we compare the empirical minimum total key dimension, $\hat{D}_K^\star$, to the predicted theoretical scaling law of $\Theta\left(\frac{m \log m}{d_{\text{model}}}\right)$, noting that optimization may fail to find a solution matching the theoretical constructive bound. And second, we test predictions regarding head count: whether a multi-head advantage appears in permutation graphs and how the empirically optimal number of heads tracks the theory.

**Experimental implementation** We empirically instantiate the idealized attention layer of our framework with two learned projections $W_Q, W_K \in \mathbb{R}^{d_{\text{model}} \times D_K}$ partitioned into $h$ heads ($d_k = D_K/h$). For a context matrix $X_\mathcal{C}$, head $k$ computes $S^{(k)} = Q^{(k)}(K^{(k)})^\top$ with $Q^{(k)} = X_\mathcal{C} W_Q^{(k)}$ and $K^{(k)} = X_\mathcal{C} W_K^{(k)}$; scores are combined by an elementwise max $S_{\max} = \max_k S^{(k)}$, and we predict an edge $(p \to q)$ iff $S_{\max}(p, q) > \tau$ for a single learned global threshold $\tau$. There is no $1/\sqrt{d_k}$ scaling, softmax, or value pathway, so capacity is purely key–query. Tasks are permutation graphs on $m$ items (one out/in-edge per node). Node embeddings $x_i \sim \mathcal{N}(0, I/d_{\text{model}})$ are $L_2$-normalized and frozen, making $D_K$ the sole capacity knob. Contexts of length $\ell$ (default $\ell=16$) are sampled with target-in-context rate $\rho=0.5$.

We train $W_Q, W_K, \tau$ with AdamW (lr $10^{-3}$, weight decay 0) using a weighted logistic loss over all ordered pairs within a context (positive weight $\ell-1$; logit sharpness $\alpha=10$), one context per step. For each run, a single permutation $\pi$ and embedding matrix are fixed by seed; training contexts are drawn on-the-fly, with 500 validation and 2,000 held-out test contexts from the same $(\ell, \rho)$ distribution. Early stopping checks validation micro-F1 every 500 steps and halts after five consecutive checks above 0.995. We report micro-F1 on the fixed test set with the single learned $\tau$; the "minimum $D_K$" is the smallest $D_K$ achieving mean test micro-F1 $\geq 0.99$ for at least one head count $h$. Full details appear in App. E.1.

### 6.1 RESULTS AND COMPARISON TO THEORY

We probe capacity on permutation graphs with $m \in \{64, 128, 256, 512\}$ and $d_{\text{model}} \in \{16, 32, 64\}$. For each $(m, d_{\text{model}})$ we sweep head counts $h \in \{1, 2, 4, 8, 16, 32, 64\}$ and several total key sizes $D_K = h\, d_k$ (multiple $D_K$ per $h$). Each configuration is trained from 10 seeds with the protocol described above (AdamW, fixed embeddings, single global threshold $\tau$). We evaluate average test micro–F1 on a fixed held-out set and define the empirical threshold

$$D_K^\star \;=\; \min\{\, D_K \;:\; \exists h \text{ s.t. mean test micro-F1} \geq 0.99 \,\}.$$

We denote by $h^\star$ a head count that attains $D_K^\star$. Full grids and per-config step limits are in App. E.2.

To isolate sequence-length effects at fixed embedding compression, we also traverse the diagonal $r \stackrel{\text{def}}{=} m/d_{\text{model}}=8$ with $(m, d_{\text{model}}) \in \{(128, 16), (256, 32), (512, 64), (1024, 128), (2048, 256), (4096, 512)\}$, using 3 seeds for the largest points and increased budgets (App. E.2). Finally, to probe extreme compression we include a second $r=32$ point $(1024, 32)$ (in addition to $(512, 16)$). We also repeat a subset of runs with a variable context length (App. E.3).

**Qualitative phenomena.** We observe: (i) a *sharp* F1 transition in a narrow $D_K$ window (capacity threshold) across all $(m, d_{\text{model}}, h)$ (Fig. 1 below and Fig. 11 in App. E.2); (ii) a pronounced *multi-head advantage* for many $(m, d_{\text{model}})$, even though each query has a single target—splitting a fixed $D_K$ across more heads reduces interference from superposition (Fig. 2); and (iii) the *optimal* head count increases with compression $r=m/d_{\text{model}}$ (Fig. 2), while per-head width at the threshold remains modest.

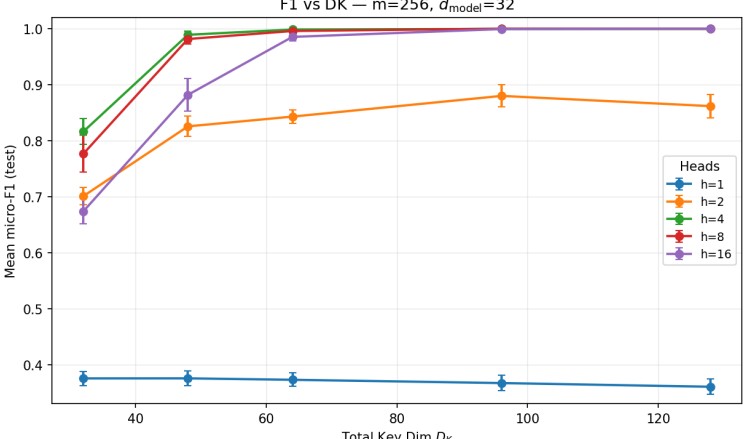

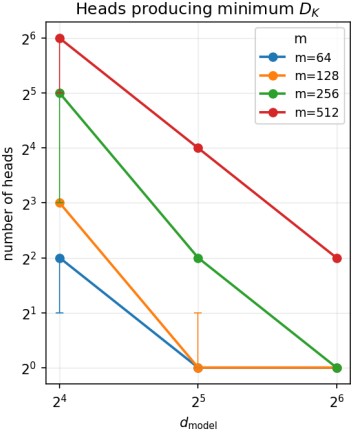

Figure 1: **Example F1–$D_K$ curve.** Each line is a fixed number of heads. A single head has significantly worse performance than multiple heads. Error bars are 95% CIs over 10 runs.

Figure 2: More heads are needed as $m$ grows and as $d_{\text{model}}$ shrinks. See App. E.2 for error bar description.

**Empirical thresholds on the base grid.** The minimum $D_K^\star$ grows rapidly with $m$ and decreases rapidly with $d_{\text{model}}$; exact values appear in Figure 10 (in the Appendix). A single head often fails to reach 0.99 F1 within the scanned $D_K$ (e.g., $(m, d_{\text{model}}) \in \{(512, 64), (256, 32)\}$, Fig. 11), whereas several small heads pass at substantially smaller $D_K$.

**Scaling laws** Plotting $D_K^\star$ against $\frac{m \log m}{d_{\text{model}}}$ yields a tight linear relation (Fig. 3):

$$D_K^\star \approx \mathbf{1.19} \cdot \frac{m \log m}{d_{\text{model}}} \qquad (R^2 = \mathbf{0.944}). \tag{4}$$

We see small deviations when $d_{\text{model}}$ is too small relative to $\log m$; this is consistent with our theoretical results. Excluding the three ($d_{\text{model}}=16$, $m>64$) points (above the line in Fig 3)—which violate the precondition $d_{\text{model}} \gtrsim c_0 \log m$ used by our constructions—gives slope 0.966 with $R^2=\mathbf{0.992}$. Thus, the empirical capacity closely matches the theoretical $\Theta\left(\frac{m \log m}{d_{\text{model}}}\right)$ rate. The head count that attains $D_K^\star$ scales approximately linearly with compression (Fig. 12 in the Appendix):

$$h^\star \approx \mathbf{1.65} \frac{m}{d_{\text{model}}} - \mathbf{6.64} \qquad (R^2 = \mathbf{0.824}). \tag{5}$$

At $D_K^\star$, per-head widths are small: $d_k^\star \in [5, 24]$ on the base grid (median 11), indicating gains come primarily from *adding heads* rather than making each head wide (Table 1; App. E.2).

**Fixed-compression diagonal** ($r=8$)**.** Holding $r$ constant collapses the prediction to $D_K^\star \propto r \log m$, so the dependence on $m$ should be logarithmic. Along $(128, 16) \to (4096, 512)$ we observe roughly this behavior from $m \geq 512$ onward (Fig. 4): $D_K^\star$ grows slowly while $m$ grows exponentially, matching the $\log m$ factor. The first two points are slightly conservative (smaller $d_{\text{model}}$) and align with the same $d_{\text{model}} \gtrsim \log m$ finite-size effect. We also expect the optimal head count to be proportional to $r$; the observed results align well with this expectation.

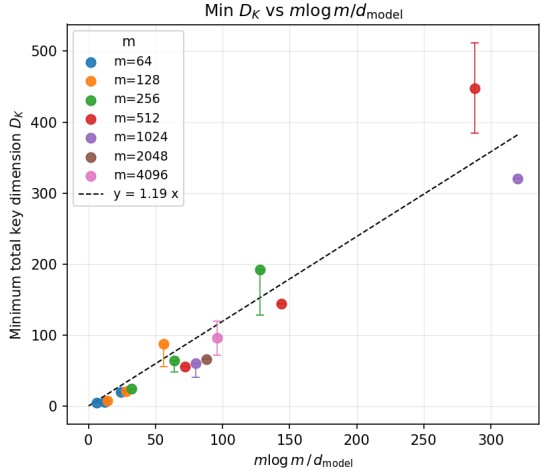

Figure 3: Comparison of $D_K^\star$ to theoretical scaling law. $x$-axis: scaling law prediction. $y$-axis: observed behavior. See App. E.2 for error bar description.

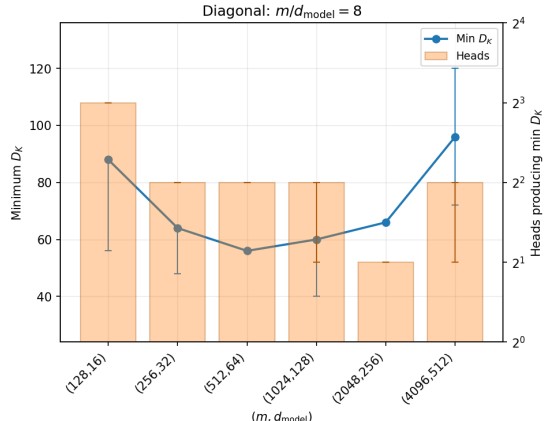

Figure 4: Fixed compression diagonal with $r = 8$. Line (left axis): minimum $D_K^\star$ achieving F1$\geq .99$. Bars (right axis): $h$ achieving that minimum. See App. E.2 for error bar description.

**Takeaways.** Empirical thresholds align closely with the $m \log m/d_{model}$ capacity law and expose a clear *multi-head advantage* even for one-target graphs. Discrepancies appear exactly where theory anticipates stronger superposition (small $d_{model}$ and very large $m$). Overall, allocating key–query budget across *more heads with modest width* is the efficient path to capacity in compressed embeddings. In Appendix E.3, provide evidence that these results are largely independent of context length. Specifically, thresholds were stable across $\ell \in \{16, 32\}$ with a small shift only when testing at longer $\ell$ than used for training.

## 7 LIMITATIONS

**Theory.** Our constructive upper bounds are tight only under a mild degree–skew assumption (e.g., $\Delta/d_{avg} \leq m/d_{model}$). When this condition is violated—or in very dense graphs—the present upper bounds do not match the information-theoretic lower bound. Moreover, our sufficiency results rely on geometric properties of the embeddings (near-orthogonality / restricted self-incoherence). While we verify these conditions for idealized random embeddings, we do not quantify how embeddings learned in trained models satisfy (or deviate from) these properties, nor the effect of such deviations on the constants in our bounds.

**Experiments.** Our search over $D_K$ is coarse because $D_K$ is tuned in multiples of the head count; there remains room to probe all cross-points in the scanned range and to explore substantially larger dimensions, but we did not do so due to compute limits. Finally, while capacity appeared largely insensitive to context length within $\ell \in \{16, 32\}$, we did not study much longer contexts for the same reason.

## 8 REPRODUCIBILITY STATEMENT

We describe the exact places in the paper and Appendix that contain the information needed for reproducibility. The task and model are formally specified in Section 3 (RGR and the QK-only attention layer), with constructive algorithms and the multi-head rationale in Sections 4–5. We provide details of our lower

bound in Appendix D, and complete proofs of our upper bounds in Appendix F. Details of the experimental setup—including synthetic data generation and context sampling, training procedure, metrics, search grids, and step budgets—is documented in Section 6 and Appendix E.1. An anonymized code package with the data generator, training/evaluation scripts, and plotting and graphing scripts is provided in the supplementary materials to reproduce all figures. These references collectively provide the assumptions, proofs, and procedures needed to re-create our results.

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

## A  EXTENSION TO SOFTMAX

We here consider a model with a more standard attention variant that applies a *softmax over the items in the context* (with the usual $d_k^{-1/2}$ scaling inside the logits). As in our base model, there is no value channel: the model produces a score per candidate softmax weight and then compares the result to a fixed threshold $\tau$. In this variant of the model, due to a lack of a value channel, we aggregate over heads via summation (rather than by taking the maximum, as is done in the base model). To see why, note that without a value channel, a head's only evidence for candidate $j$ is its softmax mass $p_{ij}^{(k)}$; any permutation-invariant linear readout across heads therefore reduces (up to scaling) to summing these masses.

**Lower bounds**  . We first point out the lower bounds for this variant of the model are unchanged from the base model. In particular, the fixed-precision bit-budget lower bound is unchanged (the parameter count is the same and the aggregator does not increase the number of labelings beyond what the parameters can encode). For the precision-agnostic bound, the change to softmax preserves the Lipschitz property of the decision function and so the covering-number/metric-entropy argument goes through with the same exponent and yields the same form of the lower bound: $\Omega\left(\frac{m'}{d_{\text{model}}} \log \frac{m^2}{m'}\right)$.

**Explicit Construction**  We next adapt Construction II (Section 4) to the softmax model. We use the same Gaussian unit-norm embedding and the same random-signature de-embedding scheme as in Algorithm 1 and Theorem 4.2. All high-probability (w.h.p.) events here mean probability at least $1 - m^{-\gamma}$ for some constant $\gamma > 2$.

Given a context $\mathcal{C} \subseteq V$ of length $\ell$, for head $k$ we form the usual (scaled) logits and per-head probabilities

$$L_{ij}^{(k)} := \frac{S_{ij}^{(k)}}{\sqrt{d_k}}, \qquad p_{ij}^{(k)} := \frac{\exp\left(L_{ij}^{(k)}\right)}{\sum_{t \in \mathcal{C}} \exp\left(L_{it}^{(k)}\right)} \quad (j \in \mathcal{C}). \tag{6}$$

The *aggregated score* we will threshold is the sum over heads

$$A_{ij} := \sum_{k=1}^{h} p_{ij}^{(k)}. \tag{7}$$

We declare that $i$ has an edge to $j$ iff $A_{ij} > \tau$ for $\tau = \frac{h-1}{\ell} + \frac{1}{2}$.

The only substantive changes from Construction II are (i) computing the per-head softmax (with scaling) over $j \in \mathcal{C}$ and (ii) replacing $\max_k$ aggregation over heads by a sum over $k$.

---

**Algorithm 2** Softmax Construction for Permutations with Compressive Embeddings

---

1: **Input:** Permutation graph $G = (V, E)$ with $\pi : V \to V$; Gaussian unit-norm embedding matrix $X \in \mathbb{R}^{m \times d_{\text{model}}}$ (rows $\mathbf{x}_i^\top$).

2: **Parameters:** $h = \frac{m}{d_{\text{model}}}$ heads; per-head width $d_k$; random Rademacher signatures as in Construction II; threshold $\tau = \frac{h-1}{\ell} + \frac{1}{2}$.

3: **Partition sources/targets:** Split $V$ into disjoint blocks $V_1, \ldots, V_h$ of size $|V_k| = d_{\text{model}}$ and $T_k := \{\pi(s) : s \in V_k\}$.

4: **Templates and de-embedding:** As in Algorithm 1, build one-hot-space templates $W'_{Q,(k)}, W'_{K,(k)}$ and set $W_Q^{(k)} = X^\top W'_{Q,(k)}, W_K^{(k)} = X^\top W'_{K,(k)}$.

5: **Per-head scores:** For each head $k$ and $(i, j)$, compute $S_{ij}^{(k)} = \langle \mathbf{q}_i^{(k)}, \mathbf{k}_j^{(k)} \rangle$, then logits $L_{ij}^{(k)} = S_{ij}^{(k)}/\sqrt{d_k}$ and softmax $p_{ij}^{(k)}$ over $j \in \mathcal{C}$.

6: **Aggregate across heads:** $A_{ij} = \sum_{k=1}^h p_{ij}^{(k)}$; declare edge $(i, j)$ present iff $A_{ij} > \tau$.

---

## A.1   MAIN GUARANTEE AND EFFICIENCY

We first state the main theorem and then prove it by reusing the score-separation from Theorem 4.2 and converting it into a probability margin under softmax.

**Theorem A.1** (Softmax analogue of Construction II). *Assume the setup of Algorithm 2. There exist absolute constants $c_0, C, C_1, C_2, \gamma > 0$ such that with $d_{model} \geq c_0 \log m$, if*

$$d_k \geq C (\log m)^2, \tag{8}$$

*then with probability at least $1 - m^{-3}$ (over the draw of $(X, \text{signatures})$), the following holds for any fixed context $\mathcal{C} \subseteq V$ of length $\ell$ that contains $\pi(i)$, simultaneously for all sources $i$ and all $j \in \mathcal{C}$:*

$$A_{i,\pi(i)} \geq 1 - m^{-\gamma} \tag{9}$$

$$A_{ij} \leq \frac{h-1}{\ell} + \Delta_\ell + m^{-\gamma} \qquad (j \neq \pi(i)), \tag{10}$$

*where, writing $\phi := C_1 \sqrt{\log m}$, we set $\Delta_\ell = C_2(e^\phi \sqrt{h \log m} + e^{2\phi} \log m)/\ell$. Consequently, $\tau = \frac{h-1}{\ell} + \frac{1}{2}$ yields $A_{i,\pi(i)} > \tau$ and $A_{ij} < \tau$ for all $j \neq \pi(i)$ whenever $\Delta_\ell \leq \frac{1}{4} - m^{-\gamma}$. A sufficient condition for this is*

$$\ell \geq 8C_2(e^\phi \sqrt{h \log m} + e^{2\phi} \log m) \ \text{with} \ \phi = C_1 \sqrt{\log m}, \tag{11}$$

*which is a lower bound for $\ell$ that is sub-polynomial in $m$. Finally, the total key dimension satisfies*

$$\boxed{D_K = h \, d_k = \Theta\Big(\frac{m}{d_{model}} (\log m)^2\Big).} \tag{12}$$

**Softmax scaling: source and necessity of the extra** $\log m$.   Relative to the max-aggregator idealization (Theorem 4.2), the per-head width $d_k$ in the softmax variant picks up an extra $\log m$ factor. This is *forced* by the standard scaling $L_{ij}^{(k)} = S_{ij}^{(k)}/\sqrt{d_k}$ inside softmax: turning an additive raw-score gap of $\Theta(d_k)$ into a softmax probability near 1 against $\ell$ distractors requires a logit gap of $\Omega(\log \ell)$, i.e. $\sqrt{d_k} = \Omega(\log \ell)$ and thus $d_k = \Omega((\log \ell)^2)$; taking the worst case $\ell \leq m$ yields $d_k = \Omega((\log m)^2)$. This lower bound is tight for our construction, which achieves $d_k = \Theta((\log m)^2)$ and hence

$$D_K = h \, d_k = \Theta\Big(\frac{m}{d_{\text{model}}} (\log m)^2\Big). \tag{13}$$

Our performance guarantee requires a context length $\ell$ of at least $h$, as well as the stated sub-polynomial in $m$ function (i.e., less than $m^\epsilon$ for any fixed $\epsilon$). We consider this reasonable, and also show empirically in Section C.2 the multi-head advantage holds even with aq very small context length of 16.

We note that removing the scaling factor $\sqrt{d_k}$ would allow us to remove this extra log factor and recover the max-aggregator's bound on $D_K$, since this would increase the logit gap on the owner head (see the proof of Theorem A.1). However, the *non-owner* heads also constrain the construction through their background mass, and as a result, the minimum requirement on $\ell$ increases from sub-polynomial in $m$ to polynomial in $M$, and in fact we require $\ell > m$. Hence, in our construction, the scaling factor is what keeps the non-owner heads' contribution flat enough for the sum-aggregator to separate.

**Remark (recovering full context-robustness).** Softmax normalization across the context prevents detecting "no target in context" without auxiliary structure (e.g., $\ell = 1$ forces $A_{ij} = h$). Adding a *single per-head null slot* (a dummy key always present) restores the analogue of Lemma 4.1: the responsible head can put mass $1 - m^{-\gamma}$ on the null when $\pi(i) \notin \mathcal{C}$, and the same $d_k = \Theta((\log m)^2)$ and $D_K = \Theta(\frac{m}{d_{\text{model}}}(\log m)^2)$ suffice; the threshold becomes $\tau = \frac{h}{\ell+1} + \frac{1}{2}$.

In summary, the extra $\log m$ factor stems from converting an additive raw-score margin into a softmax probability margin under the $d_k^{-1/2}$ scaling, and the bound $d_k = \Theta((\log m)^2)$.

## A.2 PROOF OF THEOREM A.1

We reuse the score separation from Theorem 4.2 and Lemma F.3 and then calibrate softmax probabilities.

**Step 1: Raw-score separation (from Construction II).** Let $k^\star$ be the unique head with $i \in V_{k^\star}$. The signal/noise analysis in the proof of Theorem 4.2 yields constants such that, simultaneously for all $i$ and all $j$,

$$S_{i,\pi(i)}^{(k^\star)} \geq \tfrac{3}{4} d_k, \qquad \left|S_{ij}^{(k^\star)}\right| \leq \tfrac{1}{4} d_k \quad (j \neq \pi(i)), \qquad \left|S_{ij}^{(k)}\right| \leq C_1 \sqrt{d_k \log m} \quad (k \neq k^\star). \tag{14}$$

(The first two inequalities are exactly those used in the max-aggregator proof; the last bound follows from the same leakage concentration arguments, where $N_1, N_2, N_3$ contribute only sub-Gaussian/sub-exponential noise for $k \neq k^\star$.)

Dividing by $\sqrt{d_k}$ gives the corresponding logit bounds:

$$L_{i,\pi(i)}^{(k^\star)} \geq \tfrac{3}{4}\sqrt{d_k}, \quad L_{ij}^{(k^\star)} \leq \tfrac{1}{4}\sqrt{d_k} \ (j \neq \pi(i)), \quad \left|L_{ij}^{(k)}\right| \leq C_1\sqrt{\log m} \ (k \neq k^\star). \tag{15}$$

**Step 2: Softmax calibration within the responsible head.** The following elementary lemma turns a logit gap into a softmax probability lower bound.

**Lemma A.2** (Softmax calibration). *If $z_\star \geq a$ and $z_t \leq b$ for all $t \neq \star$ in a set of size $\ell$, then*

$$\frac{e^{z_\star}}{\sum_t e^{z_t}} \geq \frac{1}{1 + (\ell - 1)e^{-(a-b)}}. \tag{16}$$

Applying Lemma A.2 to head $k^\star$ with $a = \tfrac{3}{4}\sqrt{d_k}$ and $b = \tfrac{1}{4}\sqrt{d_k}$ from (15),

$$p_{i,\pi(i)}^{(k^\star)} \geq \frac{1}{1 + (\ell - 1)e^{-\frac{1}{2}\sqrt{d_k}}} \geq 1 - m^{-\gamma}, \tag{17}$$

whenever $d_k \geq C(\log m)^2$ with $C$ large enough (since $\ell \leq m$).

**Step 3: Background mass from non-owner heads.** For any $k \neq k^\star$ and any fixed context $\mathcal{C}$, the logits $L_{it}^{(k)}$ are exchangeable across $t \in \mathcal{C}$ under the randomness of the per-head signatures (a standard symmetry trick: randomly permute the signature rows per head prior to de-embedding). Hence

$$\mathbb{E}\Big[ p_{ij}^{(k)} \,\Big|\, X, \mathcal{C} \Big] \;=\; \frac{1}{\ell} \qquad \text{for all } j \in \mathcal{C}. \tag{18}$$

Moreover, the non-owner logit bound in (15) implies a uniform *anti-spike* cap:

**Lemma A.3** (Anti-spike under bounded logits). *If $|L_{it}^{(k)}| \leq \phi$ for all $t \in \mathcal{C}$, then for every $j \in \mathcal{C}$,*

$$0 \;\leq\; p_{ij}^{(k)} \;\leq\; \frac{1}{1 + (\ell - 1)e^{-2\phi}} \;\leq\; \frac{e^{2\phi}}{\ell}. \tag{19}$$

*Proof.* The softmax at fixed $i, k$ is maximized at $j$ by taking $L_{ij}^{(k)} = +\phi$ and $L_{iz}^{(k)} = -\phi$ for all $z \neq j$, yielding $p_{ij}^{(k)} \leq \frac{e^{\phi}}{e^{\phi} + (\ell-1)e^{-\phi}} = \frac{1}{1+(\ell-1)e^{-2\phi}} \leq e^{2\phi}/\ell$. $\qquad\square$

Let $Y_k := p_{ij}^{(k)}$ for $k \neq k^\star$. Conditional on $(X, \mathcal{C})$, the variables $(Y_k)_{k \neq k^\star}$ are independent, satisfy $\mathbb{E}[Y_k] = 1/\ell$, are bounded as $0 \leq Y_k \leq b_\ell$ with $b_\ell := e^{2\phi}/\ell$, and have variance $\mathrm{Var}(Y_k) \leq \mathbb{E}[Y_k](b_\ell - \mathbb{E}[Y_k]) \leq b_\ell/\ell$. Write $S := \sum_{k \neq k^\star}(Y_k - \frac{1}{\ell})$ and $V := \sum_{k \neq k^\star} \mathrm{Var}(Y_k) \leq (h-1)b_\ell/\ell$. Bernstein's inequality (for independent, bounded summands) implies that for any $t > 0$,

$$\Pr\Big( |S| > \sqrt{2Vt} + \tfrac{b_\ell t}{3} \Big) \;\leq\; 2e^{-t}. \tag{20}$$

Using $b_\ell \leq e^{2\phi}/\ell$ we obtain the deviation (conditionally on $(X, \mathcal{C})$, hence also unconditionally)

$$\left| \sum_{k \neq k^\star} p_{ij}^{(k)} - \frac{h-1}{\ell} \right| \;\leq\; \frac{e^{\phi}\sqrt{2h\,t}}{\ell} \;+\; \frac{e^{2\phi}t}{3\,\ell} \qquad \text{with probability at least } 1 - 2e^{-t}. \tag{21}$$

A union bound over all pairs $(i, j)$ contributes a factor of at most $m^2$ events; allocating failure budget $m^{-2\gamma}$ to this step yields the choice

$$t = \log\Big(2m^{2\gamma}\Big) = O(\log m). \tag{22}$$

**Step 4: Aggregated scores and threshold.** Combining (17) and (21) for $j = \pi(i)$ yields (9). For $j \neq \pi(i)$, (21) and the owner-head wrong-mass bound from Lemma A.2 and (15) (namely $p_{ij}^{(k^\star)} \leq e^{-\frac{1}{2}\sqrt{d_k}} \leq m^{-\gamma}$) give (10). Thus, choosing $\tau = \dfrac{h-1}{\ell} + \tfrac{1}{2}$ separates $A_{i,\pi(i)}$ from all $A_{ij}$ with $j \neq \pi(i)$ whenever

$$\Delta_\ell \;=\; \frac{e^{\phi}\sqrt{2ht}}{\ell} \;+\; \frac{e^{2\phi}t}{3\,\ell} \;\leq\; \tfrac{1}{4} - m^{-\gamma}. \tag{23}$$

Imposing each term to be $\leq \frac{1}{8}$ gives the sufficient condition (11), which is satisfied for any sufficiently large $\ell$. The bound on $D_K$ is immediate from $D_K = h\,d_k$ with $h = \frac{m}{d_{\text{model}}}$ and $d_k = \Theta((\log m)^2)$. $\qquad\square$

### A.3 MULTIPLE HEADS REMAIN ADVANTAGEOUS UNDER SOFTMAX

We next show that the multi-head advantage we saw in the max over heads model variant carries over to softmax, albeit with a slightly smaller ratio. We saw above that softmax normalization introduces a

new—and unavoidable—per-head requirement: even with perfect separation of raw scores, converting an additive margin into a probability $1 - \varepsilon$ *uniformly over contexts of length $\ell$* forces $d_k = \Omega((\log \ell)^2)$ under the standard $d_k^{-1/2}$ scaling. This cost is orthogonal to the compressive de-embedding noise that drove the multi-head advantage in §5. In the softmax model, the two constraints simply stack: (i) the owner head must achieve a raw score gap large enough to survive de-embedding leakage (as in (2)–(3)), and (ii) that gap must be at least $\Omega(\sqrt{d_k})$ so that Lemma A.2 pushes the owner-head probability to $1 - \varepsilon$ against $\ell$ distractors. The first favors *more* heads (to shrink the per-head block size $B$), while the second imposes a universal $(\log \ell)^2$ floor on $d_k$.

**Proposition A.4** (Multi-head vs. single head with softmax, within the compressive template). *Under the setup of Algorithm 2 and Construction II (Gaussian unit-norm embeddings; Rademacher signatures; de-embedding noise scaling (2)), any design that succeeds w.h.p. uniformly over all contexts $\mathcal{C}$ of length $\ell \leq m$ that* contain *the true target must satisfy the combined per-head width condition*

$$d_k \ \geq \ \max\Big\{ \ C_1 (\log \ell)^2, \ \ C_2 \frac{B^2}{d_{model}^2} \log m \ \Big\}, \tag{24}$$

*where $B$ is the number of targets served by the head (the block size). Consequently:*

$$\textbf{\textit{(single head:}} \ B = m) \quad D_K^{single} \ \geq \ \Omega\Big( \frac{m^2}{d_{model}^2} \log m \ + \ (\log m)^2 \Big), \tag{25}$$

$$\textbf{\textit{(multi-head:}} \ B = d_{model}) \quad D_K^{multi} \ = \ \Theta\Big( \frac{m}{d_{model}} (\log m)^2 \Big). \tag{26}$$

*Hence, in the compressive regime $m \gg d_{model}$, the single-head total key dimension is asymptotically larger by at least*

$$\frac{D_K^{single}}{D_K^{multi}} \ \gtrsim \ \frac{(m^2/d_{model}^2) \log m}{(m/d_{model})(\log m)^2} \ = \ \frac{m/d_{model}}{\log m}. \tag{27}$$

*Proof sketch.* The leakage term $N_3(B) \asymp \frac{B}{d_{model}} \sqrt{d_k \log m}$ from (2) enforces $N_3 \leq c_1 d_k$, which is equivalent to $d_k \geq C_2 \frac{B^2}{d_{model}^2} \log m$ (Eq. (3)). This guarantees the raw owner-head separation used in Step 1 of the softmax proof (inequalities (14)–(15)). Lemma A.2 then turns an $\Omega(\sqrt{d_k})$ logit gap into owner-head mass $1 - O(m^{-\gamma})$ provided $d_k \geq C_1 (\log \ell)^2$. Summing across heads, the non-owner heads contribute a nearly uniform background $(h - 1)/\ell$ with $\Delta_\ell$ fluctuations (Eq. (21)), and Theorem A.1 separates the aggregated scores at threshold $\tau = \frac{h-1}{\ell} + \frac{1}{2}$ whenever $\ell \geq C_3 h \log m$. For the single-head case ($h = 1$) the background term vanishes, but the leakage constraint uses $B = m$, giving the stated lower bound on $d_k$; for the multi-head choice in Construction II we have $B = d_{model}$, so the leakage term becomes $O(\log m)$ and the softmax floor $C_1 (\log m)^2$ dominates, yielding the claimed $d_k^{multi}$ and $D_K^{multi}$. $\qquad \square$

## B  INCORPORATING A VALUE CHANNEL

Our QK-only model treats self-attention as *relational graph recognition* (RGR): QK decides *which connection is active*. In practice, attention also *communicates* along that connection via the value (V) channel. We here augment the softmax model of Section A with a standard value pathway, but we can also demonstrate similar results if we augmenting the base model, without softmax but with max aggregation of heads, with a value channel. Each vertex of our RGR graph now is associated with a message, and instead of just recognizing edges, the layer must retrieve all neighbors' messages. We keep the key–query (QK) mechanism, scaling, and per-head softmax unchanged.

---

**Algorithm 3** Softmax+Values for Permutations (embeddings→messages)

---

1: **Input:** Permutation graph $G = (V, E)$ with $\pi : V \to V$; structural embeddings $X \in \mathbb{R}^{m \times d_{\text{model}}}$ (rows $\mathbf{x}_i^\top$); shared value map $W_V \in \mathbb{R}^{d_{\text{model}} \times d_{\text{msg}}}$; context $\mathcal{C} \subseteq V$ of length $\ell$.

2: **Parameters (QK):** As in Alg. 2: $h = \frac{m}{d_{\text{model}}}$ heads, per-head width $d_k$, Rademacher signatures, owner partition $V = \bigsqcup_{k=1}^h V_k$ and $T_k = \{\pi(s) : s \in V_k\}$; $W_Q^{(k)} = X^\top W'_{Q,(k)}$, $W_K^{(k)} = X^\top W'_{K,(k)}$.

3: **Per-head logits and softmax:** Compute $L_{ij}^{(k)} = \langle \mathbf{q}_i^{(k)}, \mathbf{k}_j^{(k)} \rangle / \sqrt{d_k}$ and $p_{ij}^{(k)} = \text{softmax}_j(L_{ij}^{(k)})$ over $j \in \mathcal{C}$.

4: **Values and head outputs:** Set $\mathbf{v}_j^{(k)} = \mathbf{x}_j W_V$ for all $k$ and $j$. Compute $\mathbf{z}_i^{(k)} = \sum_{j \in \mathcal{C}} p_{ij}^{(k)} \mathbf{v}_j^{(k)}$.

5: **Aggregate across heads:** Output $\widehat{\mathbf{y}}_i = \sum_{k=1}^h \mathbf{z}_i^{(k)} = \sum_{j \in \mathcal{C}} A_{ij} (\mathbf{x}_j W_V)$.

---

**Messages live in the embedding span.** To avoid confounding the capacity analysis with an unrelated encoding problem, we tie messages to embeddings: each vertex $v \in V$ has an embedding $\mathbf{x}_v \in \mathbb{R}^{d_{\text{model}}}$ and a *message* defined by a shared linear map

$$\mathbf{y}_v := \mathbf{x}_v W_V \in \mathbb{R}^{d_{\text{msg}}}, \qquad W_V \in \mathbb{R}^{d_{\text{model}} \times d_{\text{msg}}}. \tag{28}$$

Thus messages lie in the linear span of the embeddings. We will take $W_V$ to be a random matrix (formalized below). This choice mirrors the Transformer value path (values are linear functions of embeddings) while keeping the focus on the capacity of attention.

Throughout this section we restrict to *permutation graphs* $G$ with a permutation $\pi : V \to V$ (each vertex has exactly one out-neighbor), as in Construction II and Theorem A.1.

**Goal (message retrieval).** Given a context $\mathcal{C} = (v_{i_1}, \ldots, v_{i_\ell})$ and a source position $i \in \mathcal{C}$, the layer should output a vector $\widehat{\mathbf{y}}_i$ that approximates the in-context neighbor's message $\mathbf{y}_{\pi(i)} = \mathbf{x}_{\pi(i)} W_V$:

$$\|\widehat{\mathbf{y}}_i - \mathbf{y}_{\pi(i)}\|_2 \text{ is small whenever } \pi(i) \in \mathcal{C}. \tag{29}$$

**Model (QK unchanged; values from embeddings).** For each head $k$, QK follow Section A:

$$\mathbf{q}_i^{(k)} = \mathbf{x}_i W_Q^{(k)}, \quad \mathbf{k}_j^{(k)} = \mathbf{x}_j W_K^{(k)}, \quad L_{ij}^{(k)} = \frac{\langle \mathbf{q}_i^{(k)}, \mathbf{k}_j^{(k)} \rangle}{\sqrt{d_k}}, \quad p_{ij}^{(k)} = \frac{e^{L_{ij}^{(k)}}}{\sum_{t \in \mathcal{C}} e^{L_{ij}^{(k)}}}. \tag{30}$$

The value path now applies the *shared* linear map $W_V$ to the embedding:

$$\mathbf{v}_j^{(k)} = \mathbf{x}_j W_V \in \mathbb{R}^{d_{\text{msg}}}, \qquad \mathbf{z}_i^{(k)} = \sum_{j \in \mathcal{C}} p_{ij}^{(k)} \mathbf{v}_j^{(k)}. \tag{31}$$

We aggregate heads by a permutation-equivariant *linear* readout that block-sums message blocks (implementable via the standard $W_O$ mechanism):

$$\widehat{\mathbf{y}}_i = \sum_{k=1}^h \mathbf{z}_i^{(k)} = \sum_{j \in \mathcal{C}} A_{ij} (\mathbf{x}_j W_V), \qquad A_{ij} := \sum_{k=1}^h p_{ij}^{(k)}. \tag{32}$$

Thus the *same* aggregated weights $A_{ij}$ that were thresholded for edge recognition now linearly mix *embedding-derived* messages.

**Random value map model.** We formalize "generic" by drawing $W_V$ at random and scaling it so that, conditional on $X$, the induced messages have isotropic covariance and sub-Gaussian tails:

**Assumption B.1** (Random value map). *$W_V$ has i.i.d. mean-zero, $\sigma/\sqrt{d_{model}}$-sub-Gaussian entries and is independent of $(X, \{W_Q^{(k)}, W_K^{(k)}\}_k)$. The embedding rows obey $\|\mathbf{x}_v\|_2 \in [c\sqrt{d_{model}}, C\sqrt{d_{model}}]$ w.h.p. for absolute constants $c, C$ (this holds for the random $X$ used in Theorem A.1). Then, conditional on $X$,*

$$\mathbb{E}[\mathbf{y}_v] = \mathbf{0}, \qquad \mathbb{E}[\mathbf{y}_v\mathbf{y}_v^\top] = \sigma^2 I_{d_{msg}}, \qquad \|\mathbf{y}_v\|_2 \le R \text{ w.h.p., with } R \lesssim \sigma\sqrt{d_{msg}}. \tag{33}$$

## B.1 MAIN GUARANTEE AND RATES

**Theorem B.2** (Softmax+Values with embedding-derived messages). *Adopt Algorithm 3 with the same QK construction and conditions as Theorem A.1: $d_{model} \ge c_0 \log m$, $h = \frac{m}{d_{model}}$ heads, and $d_k \ge C(\log m)^2$. Let $\phi = C_1\sqrt{\log m}$ and suppose Assumption B.1 holds. Then there exist absolute constants $C_*, C'_*, \gamma > 0$ such that with probability at least $1 - m^{-3}$ (over $X$, signatures, and $W_V$), the following holds for any fixed context $\mathcal{C}$ of length $\ell$ that contains $\pi(i)$, simultaneously for all sources $i \in \mathcal{C}$:*

$$\left\|\widehat{\mathbf{y}}_i - \mathbf{y}_{\pi(i)}\right\|_2 \le 2R\,m^{-\gamma} + C_*\,\sigma\,\sqrt{d_{msg}}\,\frac{(h-1)\,e^\phi}{\sqrt{\ell}} + C'_*\,R\,\frac{(h-1)\,e^{2\phi}\,\log m}{\ell}. \tag{34}$$

*Consequently, for any target $\varepsilon > 0$, it suffices to take*

$$\ell \gtrsim \frac{\sigma^2 d_{msg}}{\varepsilon^2}\,(h-1)^2 e^{2\phi} + (h-1)e^{2\phi}\log m \tag{35}$$

*to ensure $\|\widehat{\mathbf{y}}_i - \mathbf{y}_{\pi(i)}\|_2 \le \varepsilon$ for all $i$ w.h.p. The key–query budget remains*

$$\boxed{D_K = h\,d_k = \Theta\!\left(\frac{m}{d_{model}}\,(\log m)^2\right),} \tag{36}$$

*unchanged by adding values.*

Note that the value pathway uses a total value dimension of $D_V = d_{\mathrm{msg}}$ for the shared map $W_V$.

**Remark (null case).** As in the softmax remark, adding one per-head *null* key always present, with $\mathbf{v}_{\mathrm{null}} = \mathbf{0}$, restores the "no target in context" behavior: when $\pi(i) \notin \mathcal{C}$, the owner head places mass $1 - m^{-\gamma}$ on null and $\widehat{\mathbf{y}}_i \approx \mathbf{0}$ with the same rates.

## B.2 PROOF OF THEOREM B.2

We re-use the QK guarantees from Theorem A.1 and its lemmas. Fix a source $i$ and a context $\mathcal{C}$ with $\pi(i) \in \mathcal{C}$. Let $k^\star$ be the owner head ($i \in V_{k^\star}$). By eqs. (Step 2)–(Step 4) of the softmax proof,

$$p_{i,\pi(i)}^{(k^\star)} \ge 1 - m^{-\gamma}, \qquad p_{ij}^{(k^\star)} \le m^{-\gamma} \ (j \neq \pi(i)), \qquad p_{ij}^{(k)} \le \frac{e^{2\phi}}{\ell} \ (k \neq k^\star). \tag{37}$$

**Step 1 (decomposition).** Write

$$\widehat{\mathbf{y}}_i - \mathbf{y}_{\pi(i)} = \underbrace{\left(p_{i,\pi(i)}^{(k^\star)} - 1\right)\mathbf{y}_{\pi(i)}}_{\text{owner deficit}} + \underbrace{\sum_{j \neq \pi(i)} p_{ij}^{(k^\star)}\,\mathbf{y}_j}_{\text{owner spill}} + \underbrace{\sum_{k \neq k^\star}\sum_{j \in \mathcal{C}} p_{ij}^{(k)}\,\mathbf{y}_j}_{\text{non-owner background}}. \tag{38}$$

**Step 2 (owner terms).** Under Assumption B.1, $\|\mathbf{y}_j\| \leq R \lesssim \sigma\sqrt{d_{\mathrm{msg}}}$ w.h.p., so by (37),

$$\|\text{owner deficit}\|_2 \leq R\,m^{-\gamma}, \qquad \|\text{owner spill}\|_2 \leq R\,m^{-\gamma}. \tag{39}$$

**Step 3 (non-owner background under linear-span messages).** Let $w_{ij} := \sum_{k \neq k^\star} p_{ij}^{(k)}$ and $w_i := (w_{ij})_{j\in\mathcal{C}}$. Using $\mathbf{y}_j = \mathbf{x}_j W_V$,

$$\sum_{j\in\mathcal{C}} w_{ij}\,\mathbf{y}_j \;=\; \Big(\sum_{j\in\mathcal{C}} w_{ij}\,\mathbf{x}_j\Big)W_V \;=\; \underbrace{\big(X^\top w_i\big)}_{=:\;\mathbf{u}_i}\,W_V. \tag{40}$$

By (37) and Cauchy–Schwarz,

$$\|w_i\|_2^2 = \sum_{j\in\mathcal{C}} w_{ij}^2 \;\leq\; (h-1)^2\,\frac{e^{2\phi}}{\ell}. \tag{41}$$

Under the row-isotropy of $X$ in Assumption B.1, $\|\mathbf{u}_i\|_2 \leq C_X\sqrt{d_{\mathrm{model}}}\,\|w_i\|_2$ w.h.p. for an absolute constant $C_X$. Conditioned on $(X,\mathcal{C},w_i)$, $\mathbf{u}_i W_V$ is a mean-zero $\sigma$-sub-Gaussian vector with covariance $(\sigma^2\|\mathbf{u}_i\|_2^2/d_{\mathrm{model}})\,I_{d_{\mathrm{msg}}}$. A standard sub-Gaussian norm bound therefore yields, with probability at least $1 - 2e^{-t}$,

$$\Big\|\sum_{j\in\mathcal{C}} w_{ij}\,\mathbf{y}_j\Big\|_2 \;\leq\; C\,\sigma\,\sqrt{d_{\mathrm{msg}}}\,\frac{(h-1)\,e^\phi}{\sqrt{\ell}} \;+\; C'\,R\,\frac{(h-1)\,e^{2\phi}\,t}{\ell}, \tag{42}$$

and taking $t = \Theta(\log m)$ plus a union bound over $i$ produces the third term in (34).

**Step 4 (combine).** Adding the owner deficit, owner spill, and the non-owner background bounds completes (34). The sufficiency condition (35) follows exactly as before since $e^\phi = e^{C_1\sqrt{\log m}} = m^{o(1)}$. Finally, the key–query budget $D_K$ is unchanged because the QK construction and its per-head width requirement $d_k = \Theta((\log m)^2)$ are exactly those of Theorem A.1. $\qquad\square$

### B.3  MULTI-HEAD ADVANTAGE PERSISTS WITH VALUES.

The value pathway does not alter the *key–query* budget that drives the compressive advantage of multiple heads. In Alg. 3 the aggregated attention weights $A_{ij} = \sum_k p_{ij}^{(k)}$ are *exactly* those of the softmax model in §A; values only linearly mix messages using these same $A_{ij}$. Consequently, the separation requirements and the resulting $D_K$ scaling are inherited verbatim from the softmax QK analysis. As a result, we once again have that

$$\begin{aligned}
\textbf{(single head } h=1) \quad & D_K^{\mathrm{single}} \;\geq\; \Omega\Big(\frac{m^2}{d_{\mathrm{model}}^2}\,\log m \;+\; (\log\ell)^2\Big), \\[4pt]
\textbf{(multi-head } h=\tfrac{m}{d_{\mathrm{model}}}) \quad & D_K^{\mathrm{multi}} \;=\; \Theta\Big(\frac{m}{d_{\mathrm{model}}}\,(\log m)^2\Big).
\end{aligned} \tag{43}$$

Hence in the compressive regime $m \gg d_{\mathrm{model}}$ the single-head total key dimension is asymptotically larger by at least

$$\frac{D_K^{\mathrm{single}}}{D_K^{\mathrm{multi}}} \;\gtrsim\; \frac{m/d_{\mathrm{model}}}{\log m}. \tag{44}$$

# C  EXPERIMENTS WITH MODEL EXTENSIONS

We next report on experiments run in versions of the model that extend our core results. In all cases, we start with our model of permutation graphs, max over heads aggregation and no softmax, and no value channel, and then extend that model. We study the effects of denser graphs, the softmax aggregation rule, and adding a value channel.

## C.1  EXPERIMENTS FOR DENSER GRAPHS

We first consider denser graphs. Our theoretical results demonstrate tight or nearly tight asymptotic bounds for a broad class of denser graphs; this section seeks to validate and refine those results through experiments in a setting that mirrors our theoretical model. In particular, we examine $r$ regular graphs, and study the impact of scaling $r$.

Unless noted, the architecture, training loop, optimizer, early-stopping protocol, evaluation metric, and data split are identical to the permutation-graph experiments. The only substantive differences are:

- **Task / graph family.** We replace the permutation graph (out-degree $= 1$) with a directed $r$-regular graph on $m$ nodes (every node has exactly $r$ out-neighbors and $r$ in-neighbors; no self-loops, no multi-edges). Concretely, the graph is sampled as the union of $r$ random perfect matchings ("layers"), each built by a randomized greedy permutation under "forbidden" constraints to avoid self-loops and duplicates; we verify that all in-degrees equal $r$.

- **Labeling.** For a context $X_{\mathcal{C}}$, an ordered pair $(p \to q)$ is positive iff $q \in N^+(p)$ and both $p, q \in \mathcal{C}$. Thus each row can contain up to $r$ positives (vs. at most one in the permutation case). Prediction still uses the per-head scores $S^{(k)}$, an elementwise max across heads $S_{\max}$, and a single learned global threshold $\tau$.

- **Context construction (fixed target-in-context rate).** We retain the same target-in-context rate $\rho$, but now implement it over sources: we draw $b \sim \mathrm{Binom}(\ell, \rho)$ sources from the sampled context and ensure that for each selected source $u$ at least one out-neighbor of $u$ is included in the context. When $r=1$ this reduces to the permutation procedure.

- **Class-imbalance weighting.** Because the number of positives per context grows with $r$, we replace the fixed positive weight $(\ell-1)$ used for permutations with a batch-adaptive weight

$$w_+ \;=\; \frac{\#\text{negatives}}{\#\text{positives}} \tag{45}$$

  computed per context inside the same logistic loss on logits $\alpha(S_{\max} - \tau)$ (with the same $\alpha$).

Everything else (frozen, normalized Gaussian node embeddings; idealized attention with no softmax/value path; elementwise max across heads; single learned $\tau$; AdamW with the same hyperparameters; validation/test protocol; and the micro-F1 reporting criterion) is as in the baseline/core permutation-graph experiments. We also here test the specific case of $m = 128$ and $d_{\text{model}} = 32$ and scale $r$ from 1 to 32. Given the consistency of our results across different values of $m$ and $d_{\text{model}}$, we believe that this serves as a good proxy for more general behavior.

**Findings and impact (denser graphs).** Moving from permutation graphs ($r=1$) to denser, $r$-regular graphs preserves the qualitative behavior of the capacity transition and sharpens several quantitative predictions about how the key–query budget should scale.

**Sharp capacity transition persists with density.** For fixed $m$ and $d_{\text{model}}$, micro-F1 as a function of the total key budget $D_K$ remains almost step-like: performance stays low until a small window in $D_K$ where it rapidly approaches perfect recovery, and multi-head models cross the threshold at smaller $D_K$ than single-head models. This is visible for $r{=}2$ and $r{=}4$ in the F1–vs–$D_K$ sweeps in Figure 5: 1–2 heads plateau well below perfect accuracy, whereas 4–8 heads exhibit a sudden jump to micro-F1 $\approx 1$. Increasing graph density does not blur the phase transition; it simply shifts it to the right, as expected from the larger number of edges that must be separated.

**Capacity is governed by edges, not vertices.** Normalizing by the *edge* budget

$$x \;=\; \frac{m'}{d_{\text{model}}} \log_2 m \;=\; \frac{r\,m}{d_{\text{model}}} \log_2 m, \tag{46}$$

the empirically minimal $D_K$ required for high accuracy collapses to a single linear trend across degrees $r \in \{1, 2, 4, 8, 16\}$ (Figure 6). A single proportionality constant fits all densities:

$$D_K^\star \;\approx\; 0.46\,x, \tag{47}$$

so, as in our theoretical results, the total key dimension tracks $m'$ much more tightly than $m$. Intuitively, the model's *where-to-attend* budget must scale with the number of *encoded relationships*; increasing vertices without adding edges exerts far less pressure on $D_K$.

**Head count scales with edge density.** The number of heads at the empirical threshold grows with graph density and is well predicted (up to a small constant factor) by the edge-normalized ratio $m'/d_{\text{model}}$ (Figure 7). In our runs with $m{=}128$ and $d_{\text{model}}{=}32$, the head count that achieves the smallest passing $D_K$ increases roughly linearly with $r$ and stays close (within a factor of $\sim 2$) to the theoretical target $h^\star \propto m'/d_{\text{model}} = (r\,m)/d_{\text{model}}$. This reinforces a capacity-based rationale for multi-head attention: as more edges are superposed in the compressed embedding space, distributing the key–query budget across more, narrower heads reduces interference and lowers the required $D_K$.

**Relation to bounds.** For denser graphs (Figure 6, $r = 16$, the empirical thresholds lie slightly below our constructive upper-bound designs—i.e., we need slightly less total key dimension than the construction would guarantee—suggesting either slack in the analysis or other factors the model exploits during training. At the same time, as $r$ grows the gap between the constructive upper bound and the information-theoretic lower bound grows. Together, these trends indicate the true optimum is closer to the lower-bound scaling and that there is room to tighten (or redesign) dense-graph constructions.

**Takeaway.** Across densities, capacity remains a threshold phenomenon; the threshold is controlled by the *number of edges* $m'$ rather than the vertices $m$; and the optimal head count scales with $m'/d_{\text{model}}$. Practically, when budgeting attention for relational workloads, counting *relationships* is a more reliable guide than counting vocabulary size. The denser-graph experiments therefore extend the capacity law beyond permutations and provide further evidence for a principled multi-head advantage that grows with graph density.

### C.2 Experiments incorporating softmax

We also conducted experiments with the softmax version of the idealized model. These experiments mirror our experiments in the base model, with only the following changes:

- Scores are subject to a softmax along the context dimension.
- Scores are scaled by $\sqrt{d_k}$.
- Scores are aggregated over heads via a sum (instead of max).

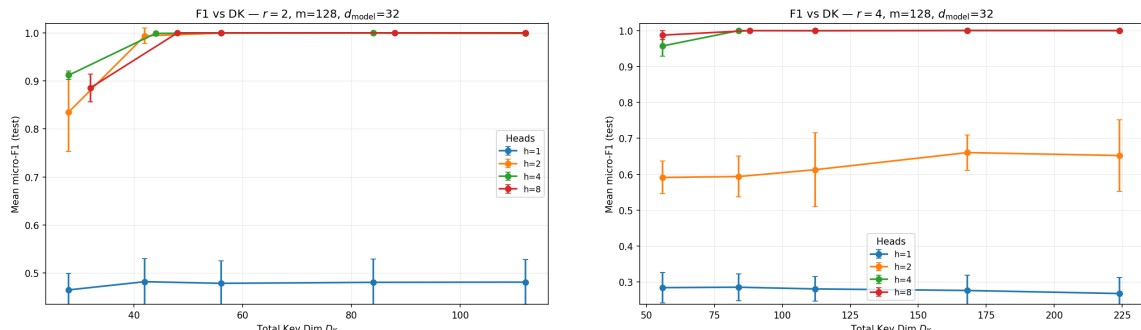

Figure 5: **Sharp transition persists and shifts right for** $r \in \{2, 4\}$**.** Mean test micro-F1 vs. total key dimension $D_K$ for $m=128$, $d_{\text{model}}=32$. More heads reach perfect recovery at smaller $D_K$; for $r = 2$, a single head plateaus well below 1.0, while for $r = 4$ 2 heads has the same property. More heads exhibit a sudden jump to $\approx 1.0$.

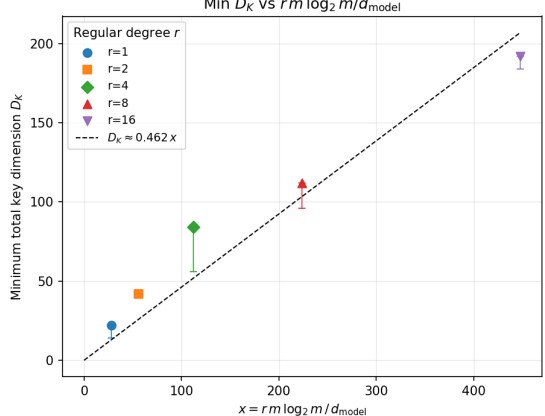

Figure 6: **Capacity scales with** $m'$**.** Minimal passing $D_K$ versus $x = (r\, m/d_{\text{model}}) \log_2 m$ collapses across degrees $r$. The dashed line $D_K \approx 0.46\, x$ is a one-parameter fit, highlighting that the number of *edges* $m'$ predicts the threshold more accurately than the vocabulary size $m$. Error bars are calculated using the same methodology as described in Appendix E.2.

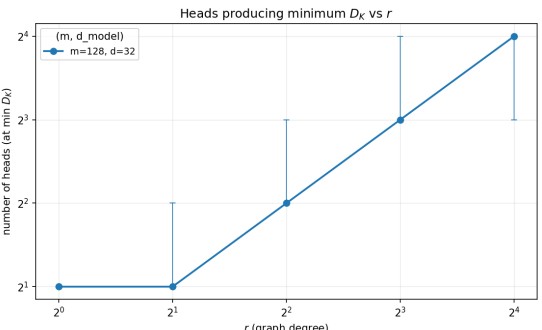

Figure 7: **Optimal head count rises with density.** Number of heads that achieves the smallest passing $D_K$ versus graph degree $r$ (here $m=128$, $d_{\text{model}}=32$). The trend tracks $m'/d_{\text{model}} = (r\, m)/d_{\text{model}}$ up to a constant factor and shows increasing benefit from additional heads as the graph becomes denser. Error bars indicate neighboring head counts that tied for the minimum within the measurement resolution (using the same methodology as described in Appendix E.2).

• Training was allowed to run for up to 80,000 steps.

We depict several examples of the resulting F1-$D_K$ curves in Figure 8. As in the model with max over heads (and no softmax), we here see a distinct multi-head advantage when in the compressed embedding range of $m \gg d_{\text{model}}$. We also see that the required $D_K$ to reach an F1 of 0.99 is shifted to the right from the experiments run in our core model - consistent with the additional $\log m$ factor in our constructions.

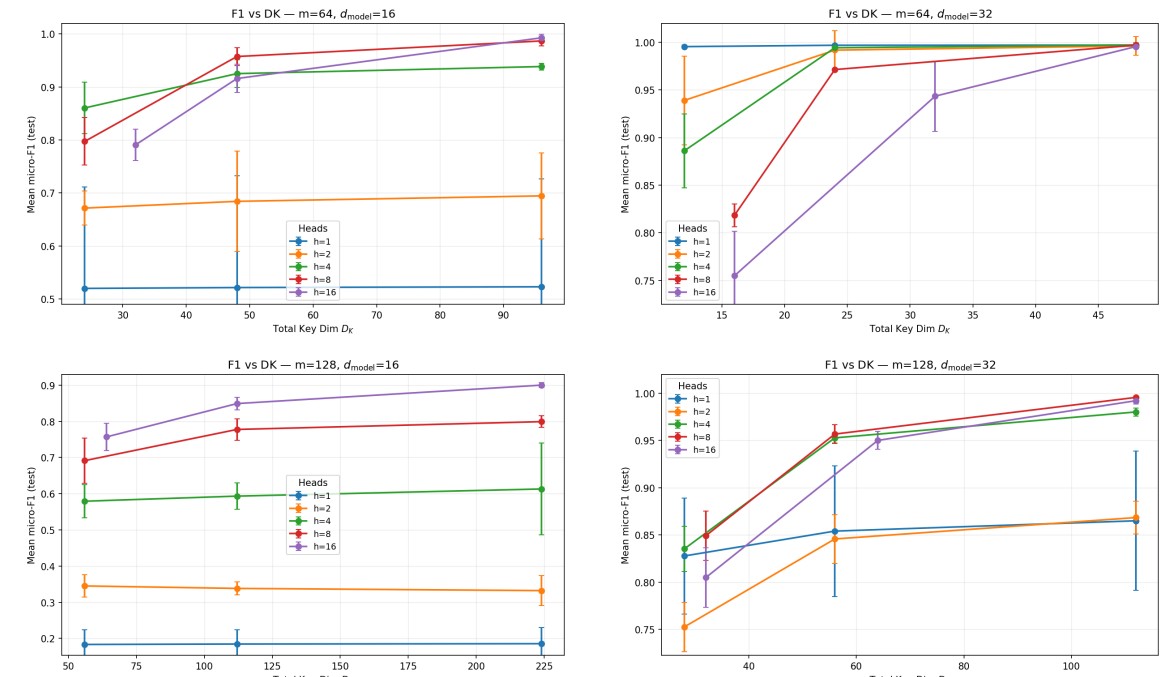

Figure 8: **Example F1–$D_K$ curves with softmax incorporated.** Each panel fixes $(m, d_{\text{model}})$ and sweeps heads $h$ and $D_K = h\, d_k$; markers show mean test micro–F1 and error bars are 95% CIs over 3 runs. We see a distinct multi-head advantage when in the compressed embedding regime - i.e., all cases except $m = 64$, $d_{\text{model}} = 32$.

## C.3  EXTENSION TO A VALUE CHANNEL

We next study the empirical impact of adding a value channel to our model. All aspects of the data generation and optimization protocol remain in our core model (permutation graphs on $m$ nodes, frozen normalized embeddings $x_i \in \mathbb{R}^{d_{\text{model}}}$, contexts of length $\ell$ with target-in-context rate $\rho$, and AdamW with the same learning rate and regularization), except that we equip the layer with a value pathway and change the learning objective from edge classification to message retrieval.

Concretely, the key–query path is unchanged: we retain the same learned projections $W_Q, W_K \in \mathbb{R}^{d_{\text{model}} \times D_K}$, partitioned into $h$ heads with per-head width $d_k = D_K/h$, and we continue to aggregate scores by an elementwise max over heads $S_{\max} = \max_k Q^{(k)}(K^{(k)})^\top$ without $1/\sqrt{d_k}$ scaling or softmax. The only architectural addition is a *random value map* $W_V \in \mathbb{R}^{d_{\text{model}} \times d_{\text{msg}}}$, drawn once at initialization and then frozen. Each node $i$ carries a "message" $y_i = x_i W_V \in \mathbb{R}^{d_{\text{msg}}}$ that lies in the span of the original embeddings, matching the theoretical assumption of a fixed random value channel.

Given a context $\mathcal{C}$ with embedding matrix $X_{\mathcal{C}}$, we reuse the same attention scores $S_{\max} \in \mathbb{R}^{\ell \times \ell}$ to mix value messages. Let $V_{\mathcal{C}} = X_{\mathcal{C}} W_V$ collect the in-context messages. For each position $i$ whose outgoing permutation neighbor $\pi(i)$ also appears in the context, we form a predicted neighbor message

$$\hat{y}_i = \left(S_{\max} V_{\mathcal{C}}\right)_i \tag{48}$$

using the raw (un-normalized) scores in $S_{\max}$ as mixing weights, and we supervise it toward the true neighbor message $y_{\pi(i)} = x_{\pi(i)} W_V$. Training minimizes the mean squared error between $\hat{y}_i$ and $y_{\pi(i)}$ over all such "valid" positions in the batch, i.e.,

$$\mathcal{L}_{\text{MSE}} \;=\; \frac{1}{|\mathcal{I}_{\text{valid}}|} \sum_{i \in \mathcal{I}_{\text{valid}}} \left\| \hat{y}_i - y_{\pi(i)} \right\|_2^2, \tag{49}$$

where $\mathcal{I}_{\text{valid}}$ is the set of indices whose permutation neighbor lies in the same context. Early stopping now operate on validation MSE for this message-retrieval task (with the same check interval and patience as before), and we report test-set MSE as the primary metric. Micro-F1 and score margins induced by the learned $W_Q, W_K$ no longer play any role in optimization or stopping.

**Findings (Value Retrieval).** We observe that the multi-head advantage found with edge classification transfers directly to the message-retrieval task. Figure 9 displays the test MSE as a function of total key dimension $D_K$ for $m = 64$ with embedding dimensions $d_{\text{model}} = 16$ and $d_{\text{model}} = 32$.

**Multi-head advantage in compressed regimes.** In the compressed regime where $d_{\text{model}} \ll m$ (Figure 9, left, $d_{\text{model}} = 16$), we observe a significant separation between single-head and multi-head performance. The single-head model ($h = 1$) fails to reduce MSE significantly even as $D_K$ increases, plateauing at a high error rate. In contrast, models with $h \geq 4$ are able to leverage the key budget effectively, driving the MSE down sharply. This confirms that even when the task is soft message retrieval rather than hard binary classification, the geometric bottleneck of identifying the correct neighbor requires the noise suppression impact of multiple heads.

**Behavior in less compressed regimes.** When the embedding dimension is relaxed to $d_{\text{model}} = 32$ (Figure 9, right), the necessity for multiple heads diminishes, consistent with our theoretical predictions. Here, a single head ($h = 1$) is competitive, and in some low-$D_K$ settings even outperforms highly fragmented architectures (e.g., $h = 16$, where $d_k$ becomes very small). However, intermediate head counts ($h = 4, 8$) still achieve the lowest ultimate MSE.

**Takeaway.** These results demonstrate that our core findings are not an artifact of the thresholded classification objective. The attention mechanism's ability to route information—specifically, to select the correct value vector to mix—is governed by the same geometric capacity constraints derived for the edge-existence problem.

## D  LOWER BOUND ON RELATIONAL GRAPH RECOGNITION

In this section, we provide our lower bound on any self-attention mechanism that uniformly recovers every directed graph on $m$ items with exactly $m'$ edges in our model from Section 3 under a fixed positive margin $\gamma$. The number of such graphs is $\binom{m(m-1)}{m'}$; essentially what we show is that the QK parameters must carry (at least) the description length of the edge set, and thus the total key dimension

$$D_K \;=\; \Omega\!\left( \frac{\log \binom{m(m-1)}{m'}}{d_{\text{model}}} \right) = \Omega\!\left( \frac{m' \log(m^2/m')}{d_{\text{model}}} \right). \tag{50}$$

The result is independent of parameter precision and applies to any context length $\ell \geq 2$.

We start with some preliminaries. We first point out that we can focus only on length-2 contexts. Uniform correctness for RGR requires that, for *every* ordered pair $(u, v) \in V \times V$, the decision "$(u, v) \in E$?" is the same in every context containing $u$ and $v$. In particular it must be correct in the length-2 context $\mathcal{C} = (u, v)$.

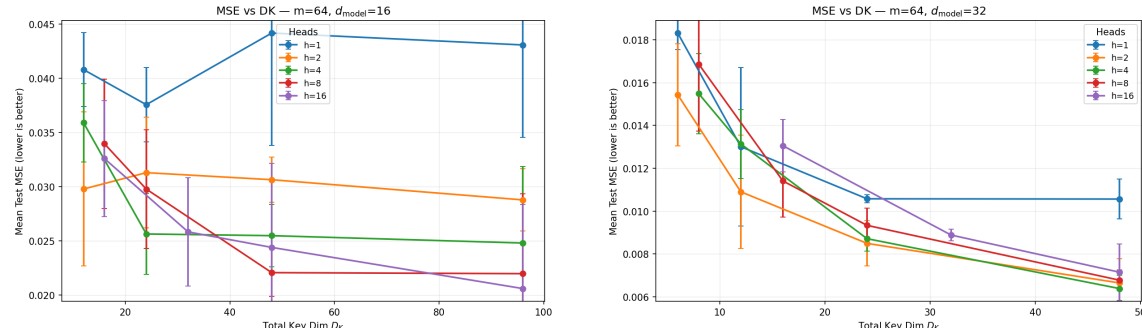

Figure 9: **Value retrieval MSE mirrors capacity transitions.** Mean test MSE vs. total key dimension $D_K$ for a value-retrieval task with frozen random value mappings ($m{=}64$). **Left ($d_{\text{model}}{=}16$):** In the compressed regime, single-head attention (blue) fails to retrieve the correct message, while multi-head attention succeeds. **Right ($d_{\text{model}}{=}32$):** With less compression pressure, the gap narrows, though multi-head models still attain the lowest final MSE. Error bars are 95% CIs over 3 runs

Hence any lower bound proved using only length-2 contexts applies to the full problem. We next provide two structural reductions.

(i) *Multi-head to a single bilinear form.* For a length-2 context and any monotone per-pair head aggregator (max, log-sum-exp, sum), replacing it by the *sum* only makes the model more permissive. Writing

$$M \;=\; \sum_{k=1}^{h} W_Q^{(k)} W_K^{(k)\top} \;\in\; \mathbb{R}^{d_{\text{model}} \times d_{\text{model}}}, \tag{51}$$

the score on $(u, v)$ is the single bilinear form

$$S(u, v) \;=\; \mathbf{x}_u^\top M \, \mathbf{x}_v, \tag{52}$$

and an edge is recognized iff $S(u, v) > \tau$ for a global threshold $\tau$. Since each summand $W_Q^{(k)} W_K^{(k)\top}$ has rank at most $d_k$,

$$\text{rank}(M) \;\leq\; \sum_{k=1}^{h} d_k \;=\; D_K. \tag{53}$$

Thus, on length-2 contexts, multi-head QK collapses to thresholding a *single rank-$\leq D_K$* bilinear form.

(ii) *Edge-set description length.* Let $m = |V|$ and $N = m(m - 1)$ be the number of ordered, loop-free pairs. For any target size $m'$, the family $\mathcal{G}_{m,m'}$ of directed graphs with exactly $m'$ edges has cardinality $\binom{N}{m'}$; hence any procedure that can realize an arbitrary $E \subseteq [N]$ with $|E| = m'$ must (implicitly) transmit at least

$$L(m, m') \;=\; \ln \binom{N}{m'} \tag{54}$$

nats about the edge set. Our proof quantifies the number of distinct edge labelings the rank-$\leq D_K$ bilinear model can realize at margin $\gamma$, via a covering-number (metric entropy) argument, and compares it to $\binom{N}{m'}$.

**Definition D.1** (Constant-margin recovery on length-2 contexts). A parameter choice $\left(\{W_Q^{(k)}, W_K^{(k)}\}_{k=1}^{h}, \tau\right)$ *recovers* a graph $G = (V, E)$ with margin $\gamma > 0$ if, for every ordered pair $(u, v)$ with $u \neq v$,

$$(u, v) \in E \Rightarrow \mathbf{x}_u^\top M \, \mathbf{x}_v \geq \tau + \gamma, \qquad (u, v) \notin E \Rightarrow \mathbf{x}_u^\top M \, \mathbf{x}_v \leq \tau - \gamma, \tag{55}$$

where $M = \sum_k W_Q^{(k)} W_K^{(k)\top}$.

Because the decision rule is homogeneous in $(M, \tau)$, we fix scale by requiring

$$M = UV^\top, \quad \|U\|_F \le 1, \ \|V\|_F \le 1, \ |\tau| \le 1. \tag{56}$$

Under (56) and $\|\mathbf{x}_u\|_2, \|\mathbf{x}_v\|_2 \le 1$, the score map is 1-Lipschitz in Frobenius norm:

$$\left| \mathbf{x}_u^\top UV^\top \mathbf{x}_v - \mathbf{x}_u^\top \tilde{U}\tilde{V}^\top \mathbf{x}_v \right| \ \le \ \|U - \tilde{U}\|_F + \|V - \tilde{V}\|_F, \qquad \text{and } |(\tau - \tilde{\tau})| \text{ adds linearly.} \tag{57}$$

Therefore any perturbation of $(U, V, \tau)$ of radius at most $\gamma/4$ preserves all pairwise signs, and hence the entire edge set, on length-2 contexts.

**Theorem D.2** (Description-length lower bound for QK). *Fix $m \in \mathbb{N}$ and $m' \in \{0, \ldots, m(m-1)\}$. Suppose an attention mechanism of the form* (52)–(53)*, with item embeddings $\|\mathbf{x}_v\|_2 \le 1$, can recover every graph in $\mathcal{G}_{m,m'}$ with margin $\gamma \in (0, 1)$. Then there exists a constant $c(\gamma) > 0$ such that*

$$d_{model} \, D_K \ \ge \ c(\gamma) \log \binom{m(m-1)}{m'} \ - \ O(1). \tag{58}$$

*Equivalently,*

$$D_K \ = \ \Omega\!\left( \frac{\log \binom{m(m-1)}{m'}}{d_{model}} \right). \tag{59}$$

*Proof.* Under the normalization (56) and Lipschitz property (57), the parameter set $\mathbb{B} = \{(U, V, \tau) : \|U\|_F, \|V\|_F, |\tau| \le 1\}$ admits an $\varepsilon$-net of radius $\varepsilon = \gamma/4$ of size at most

$$N_{\mathrm{cov}}(\varepsilon) \ \le \ \left( \frac{C}{\varepsilon} \right)^{2 \, d_{\mathrm{model}} D_K + 1} \ = \ \left( \frac{C'}{\gamma} \right)^{2 \, d_{\mathrm{model}} D_K + 1} \tag{60}$$

for absolute constants $C, C' > 0$. Each net point induces a *unique* labeling of the $N = m(m-1)$ ordered pairs by the margin, hence at most $N_{\mathrm{cov}}(\varepsilon)$ distinct edge sets can be realized. Since the mechanism must realize all $\binom{N}{m'}$ edge sets of size $m'$, we obtain $\binom{N}{m'} \le N_{\mathrm{cov}}(\varepsilon)$, which rearranges to (58)–(59). $\qquad \square$

**Equivalent finite-precision statement.** If each real parameter in $\{W_Q^{(k)}, W_K^{(k)}, \tau\}$ has $b = \Theta(1)$ *effective bits* after normalization (e.g., due to quantization or stochastic rounding), then the parameter budget contains at most $B = b \, (2 \, d_{\mathrm{model}} D_K + 1)$ bits and thus can realize at most $2^B$ distinct edge sets. Requiring $2^B \ge \binom{N}{m'}$ gives the same conclusion as (59) with an explicit constant $1/(2b)$. Under the margin model above, one may take $b = \Theta(\log(1/\gamma))$.

**Bounds for specific cases.** This demonstrates the following results:

- Exactly $m' = m$ edges (such as permutation graphs): $D_K = \Omega\!\left( \frac{m' \log m}{d_{\mathrm{model}}} \right)$.

- Dense regime with $m' = \Theta(m^2)$: $D_K = \Omega\!\left( \frac{m'}{d_{\mathrm{model}}} \right)$.

- Sparse regime with $m' = O(m^{2-\epsilon})$ for some positive constant $\epsilon$: $D_K = \Omega\!\left( \frac{m' \log m}{d_{\mathrm{model}}} \right)$.

# E   MORE DETAILS ON OUR EXPERIMENTS

## E.1   EXPERIMENTAL IMPLEMENTATION DETAILS

This appendix reproduces the full experimental protocol (model specification, context sampling procedure, loss, optimization, early stopping, and evaluation criteria) described in Section 6.

Our experiments instantiate the upper-bound model from Section 3 as follows. The parameters are two learned projections $W_Q, W_K \in \mathbb{R}^{d_{\text{model}} \times D_K}$ and a *single global scalar threshold* $\tau$. We conceptualize $W_Q, W_K$ as $h$ head blocks of width $d_k = D_K/h$. Scoring is done for a context matrix $X_{\mathcal{C}} \in \mathbb{R}^{\ell \times d_{\text{model}}}$, where head $k$ produces $S^{(k)} = Q^{(k)}(K^{(k)})^\top \in \mathbb{R}^{\ell \times \ell}$ with $Q^{(k)} = X_{\mathcal{C}} W_Q^{(k)}$ and $K^{(k)} = X_{\mathcal{C}} W_K^{(k)}$. Scores are aggregated by elementwise max across heads: $S_{\max} = \max_k S^{(k)}$. At evaluation time we predict an edge $(p \to q)$ iff $S_{\max}(p, q) > \tau$. This matches the theoretical mechanism exactly: there is *no* $1/\sqrt{d_k}$ scaling, *no* softmax, and *no* value pathway—so capacity is purely key–query driven.

Our primary testbed is the family of permutation graphs $\{(V, E_\pi)\}$ with $|V| = m$ and $E_\pi = \{(i, \pi(i)) : i \in V\}$, where $\pi$ is a uniformly random permutation. This realizes the $m' = m$ constructive case used in our upper bound and isolates the single-target setting in which head specialization is most interpretable. Consistent with our constructions, each node $i \in V$ has a fixed embedding $x_i \in \mathbb{R}^{d_{\text{model}}}$ drawn i.i.d. from $\mathcal{N}(0, I/d_{\text{model}})$ and then $L_2$-normalized. Embeddings are frozen throughout training and evaluation. This both aligns with the random (nearly orthogonal) embedding assumption in our proofs and makes $D_K$ the sole capacity knob. An example is a context $\mathcal{C}$ of length $\ell$ (baseline $\ell = 16$). To prevent degenerate class imbalance when $\pi(i)$ often falls outside $\mathcal{C}$, we enforce a target per-context positive rate $\rho \in (0, 1)$ as follows:

1. Sample a set $S \subseteq V$ of $\ell$ distinct nodes uniformly.
2. Sample $b \sim \text{Binomial}(\ell, \rho)$ and choose $U \subseteq S$ with $|U| = b$.
3. For each $i \in U$, if $\pi(i) \notin S$ replace a random $j \in S \setminus \{i\}$ by $\pi(i)$, preserving $|S| = \ell$ and distinctness.

All of our experiments use $\rho = 0.5$. This preserves the RGR semantics (positives remain exactly those $(i, \pi(i))$ that land in the same context) while reducing the time required to train.

For each experiment we sample one permutation $\pi$ and one embedding matrix $X$ using a fixed seed. We then generate a validation set of 500 contexts and a held-out test set of 2,000 contexts with the same $(\ell, \rho)$ distribution. We then draw training contexts on the fly from the same generator (one context per optimization step).

We train $W_Q, W_K, \tau$ by minimizing a weighted logistic loss on all ordered pairs within a context:

$$z_{pq} = \alpha\big(S_{\max}(p, q) - \tau\big), \quad \mathcal{L} = \frac{1}{|\mathcal{C}|^2} \sum_{p,q} \Big[ \underbrace{\text{softplus}(-z_{pq}) y_{pq}}_{\text{positive term}} \cdot \underbrace{\text{pos}_{\text{weight}}}_{=\ell-1} + \text{softplus}(z_{pq})(1 - y_{pq}) \Big], \quad (61)$$

where $y_{pq} = 1$ iff $(v_{i_p}, v_{i_q}) \in E$. The weighting $\text{pos}_{\text{weight}} = \ell - 1$ reflects that each source has at most one positive among $\ell$ candidates.

We use AdamW with learning rate $10^{-3}$ and weight decay 0. Parameters are initialized with $W_Q, W_K \sim \mathcal{N}(0, 1/\sqrt{d_{\text{model}}})$ and $\tau = 0$. The logit sharpness is $\alpha = 10$. We train for a number of steps with early stopping: every 500 steps we compute validation micro-F1; if it exceeds 0.995 for 5 consecutive checks, training halts. The number of steps increases with problem complexity. We use one context per step (contexts are small and independent), which keeps the implementation close to the theoretical algorithm and avoids artifacts from large mini-batches.

All evaluation is conducted on the fixed held-out test set of 2,000 contexts using the *single learned* threshold $\tau$ shared across all contexts. Our metric is *Micro-F1* over all ordered pairs across all test contexts. This directly measures correctness of binary edge recognition per the RGR objective. While the *stopping rule* uses validation F1 > 0.995, the *minimum* $D_K$ we report below is extracted on the *test* set using a looser criterion: the smallest $D_K$ achieving mean micro–F1 $\geq 0.99$ for at least one $h$. We use 0.99 to keep a margin from the stopping rule. All tables and statements about minimum $D_K$ are based on this 0.99 test criterion.

### E.2 Result details

We provide more detail on the results we found, additional details on the configurations used to find them, as well as the methodology we used for error bar determination.

#### Methodology for Error Interval Construction

**Display CIs for F1 curves.** Unless otherwise noted, error bars are 95% $t$-intervals across seeds: $\bar{F}_1 \pm t_{0.975,\,n-1}\,s/\sqrt{n}$, where $n$ is the number of runs and $s$ their sample standard deviation. Intervals reflect training-run variability with a fixed test set.

**Minimum key dimension $D_K^\star$.** The error interval for the minimum total key dimension, $D_K^\star$, is designed to reflect the uncertainty in the F1 score. For any given model configuration, we determine a central estimate along with an optimistic lower bound and a conservative upper bound, all based on a required F1 score of at least 0.99.

Let the mean F1 score from a set of trials be $\bar{F}_1$, with its corresponding 95% confidence interval being $[F_{1,\text{low}}, F_{1,\text{high}}]$. The three reported values for $D_K^\star$ are defined as follows:

- **Central Estimate:** The primary value reported. It's the minimum $D_K$ found for which the **mean F1 score** meets the performance threshold ($\bar{F}_1 \geq 0.99$).

- **Conservative Upper Bound:** This is the minimum $D_K$ for which the **lower bound of the F1 confidence interval** meets the threshold ($F_{1,\text{low}} \geq 0.99$). This stricter condition identifies the $D_K$ needed to be 95% confident that the true performance is sufficient.

- **Optimistic Lower Bound:** This is the minimum $D_K$ for which the **upper bound of the F1 confidence interval** meets the threshold ($F_{1,\text{high}} \geq 0.99$). This looser condition identifies the $D_K$ for which it is merely plausible that the true performance is sufficient.

**Optimal number of heads.** Let $h^\star$ be the head count achieving $D_K^\star$ (ties broken by larger $\bar{F}_1$). We form a candidate pool of head counts whose tested $D_K$ lies within 10% of $D_K^\star$. Each candidate is compared to $h^\star$ using a *paired* two-sided $t$-test on per-seed F1; candidates with $p > 0.05$ are labeled "not significantly different" and retained.[4] The reported interval spans the minimum and maximum head counts retained.

#### Results

Figure 10 lists $D_K^\star$, the minimum total key dimension, found for each configuration of $m$ and $d_{\text{model}}$ we tested. We use our minimum key dimension $D_K^\star$ intervals methodology, with the upper right corner being the upper bound and the lower left corner being the lower bound. The $d_K^\star$ (per head key size) used to achieve these $D_K^\star$s are shown in Table 1, for numbers in the main sweep.

---

[4]We do not interpret $p > 0.05$ as proof of equivalence; it only indicates insufficient evidence of a difference at $\alpha = 0.05$.

Minimum DK achieved

| $m$ \ $d_{\text{model}}$ | 16 | 32 | 64 | 128 | 256 | 512 |
|---|---|---|---|---|---|---|
| 64 | **20** (18–20) | **6** (6–6) | **5** (5–5) | — | — | — |
| 128 | **88** (56–88) | **21** (21–21) | **7** (7–7) | — | — | — |
| 256 | **192** (128–192) | **64** (48–64) | **24** (24–24) | — | — | — |
| 512 | **448** (384–704) | **144** (144–144) | **56** (56–56) | — | — | — |
| 1024 | — | **320** (320–320) | — | **60** (40–60) | — | — |
| 2048 | — | — | — | — | **66** (66–66) | — |
| 4096 | — | — | — | — | — | **72** (144–192) |

Figure 10: **Minimum total key dimension** $D_K^\star$. Upper right and lower left numbers represent confidence range; methodology described in the text.

|  | $d_{\text{model}}$ | | |
|---|---|---|---|
| $m$ | 16 | 32 | 64 |
| 64 | 5 | 6 | 5 |
| 128 | 11 | 21 | 7 |
| 256 | 6 | 16 | 24 |
| 512 | 7 | 9 | 14 |

Table 1: **Per-head key dimension** $d_k$ from the main sweep.

These are found using the training step upper bounds shown in Table 2, where we increase the steps as the problem size and complexity increases.

Also, we provide additional examples of our findings from the main sweep of configurations in Fig. 11.

In Fig. 12, we plot the number of heads used in the optimal found configuration versus the compression $m/d_{\text{model}}$.

### E.3 SENSITIVITY TO CONTEXT LENGTH.

To probe whether capacity depends on the context length $\ell$, we repeated the $D_K$ sweep with $h = 8$ for three train/test settings: $(\ell_{\text{train}}, \ell_{\text{test}}) \in \{(16, 16), (16, 32), (32, 32)\}$ (Figure 13). Across all $(m, d_{\text{model}})$ pairs the F1–$D_K$ curves are strikingly similar: the sharp transition and the minimum $D_K$ at which each configuration "passes" shift only slightly with $\ell$. Two small, consistent effects are visible: (i) *longer test*

| $m$ | $d_{\text{model}}$ | Training step cutoff |
|---|---|---|
| 64 | 16, 32, 64 | 20,000 |
| 128 | 16, 32, 64 | 20,000 |
| 256 | 32, 64 | 20,000 |
| 256 | 16 | 30,000 |
| 512 | 32, 64 | 20,000 |
| 512 | 16 | 80,000 |
| 1024 | 128 | 80,000 |
| 2048 | 256 | 80,000 |
| 4096 | 512 | 200,000 |

Table 2: **Training step cutoffs by configuration.** Default cutoff is 20,000 steps, with extended budgets for larger problem sizes.

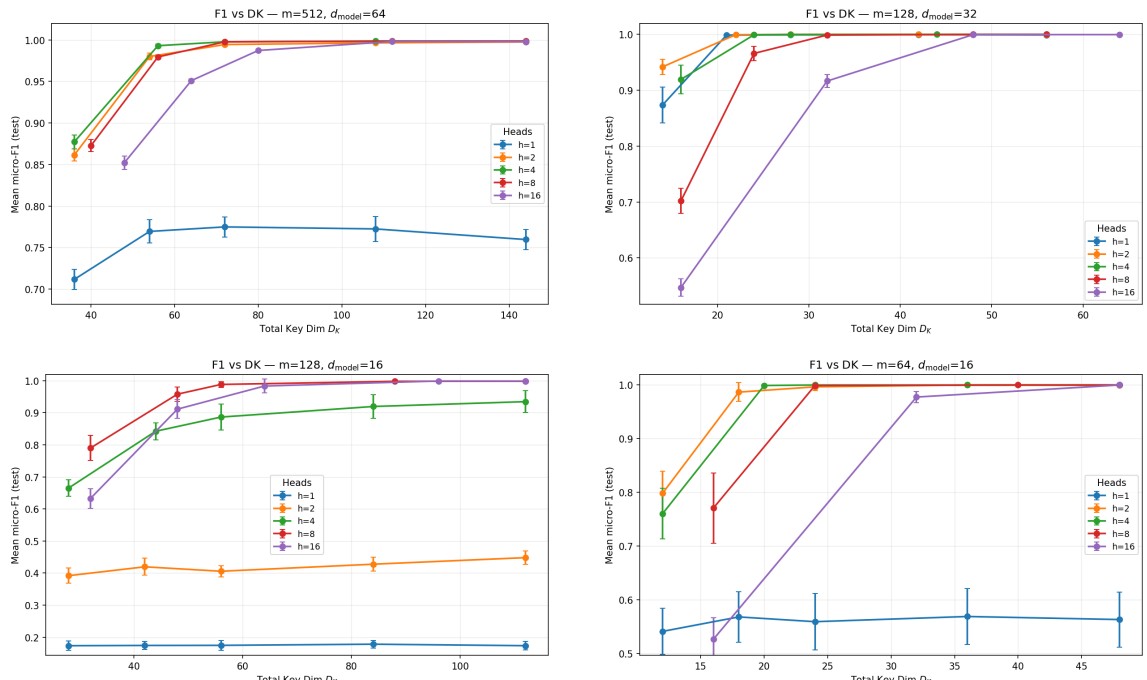

Figure 11: **Example F1–$D_K$ curves.** Each panel fixes $(m, d_{\text{model}})$ and sweeps heads $h$ and $D_K = h\, d_k$; markers show mean test micro–F1 and error bars are 95% CIs over 10 runs. The transition from failure to success occurs at a configuration-specific $D_K$ threshold which is dependent on $h$.

*contexts without retraining* (16↦32) incur a modest right-shift and/or reduced saturation, most noticeably in the most compressed regime (e.g., $m = 256, d_{\text{model}} = 16$). This is expected because our metric is micro-F1 over all ordered pairs: with $\rho = 0.5$ the positive fraction is $\rho/\ell$, so doubling $\ell$ halves the base rate while the single global threshold $\tau$ learned at $\ell = 16$ remains fixed. (ii) *retraining at the longer length* (32↦32) largely closes that gap, bringing the curves back in line with the 16↦16 condition. Overall, the empirical capacity threshold is governed primarily by $(m, d_{\text{model}})$ and only weakly by $\ell$ over the range we tested; when

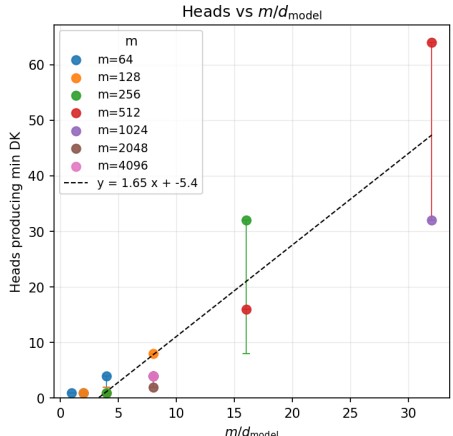

Figure 12: The number of heads needed grows approximately linearly with compression; the dashed line shows a least-squares fit. See text for a description of the error bars. We do not tie the line to the origin, since heads are clipped at $h \geq 1$.

test-time contexts are longer than those seen in training, a small increase in $D_K$ or simply training at the longer length suffices to recover performance.

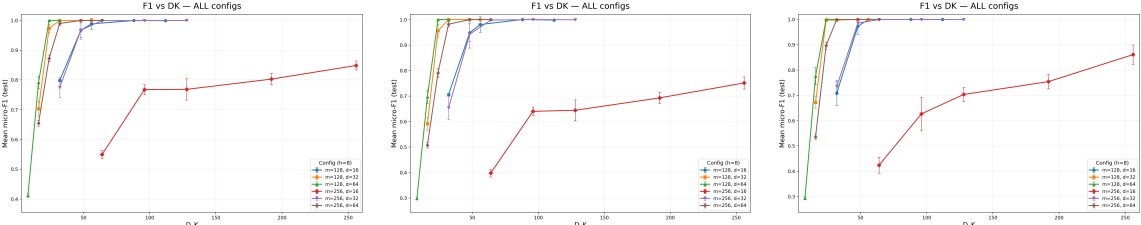

Figure 13: **Effect of context length on F1–$D_K$.** Left: train/test $\ell = 16/16$; middle: 16/32 (longer contexts only at test time); right: 32/32. Error bars are 95% CI over three seeds. Curves and thresholds are nearly length-invariant; the only systematic drop occurs when evaluating at longer $\ell$ without retraining, which is largely removed by training at the longer length.

## F   FURTHER DETAILS ON OUR EXPLICIT CONSTRUCTIONS

We here provide additional details on our explicit constructions from Section 4. We start with the easiest case - permutation graphs with no embedding. This result is subsummed by the second construction, so is only included as a warm up for the more general case.

### F.1   CONSTRUCTION I: PERMUTATION GRAPHS WITH ONE-HOT EMBEDDINGS

**Setup.** We start with the following two assumptions: (i) $G$ is a permutation on $m$ items, defined by a function $\pi : V \to V$, where the edges are $E = \{(v_i, v_{\pi(i)}) \mid v_i \in V\}$, and thus $m' = m$. (ii) Node $v_i$ is represented by the one-hot vector $\mathbf{x}_i = \mathbf{e}_i \in \mathbb{R}^m$, setting the model dimension $d_{\text{model}} = m$.

With these assumptions, a single attention head ($h = 1$) suffices, so the total key dimension is $D_K = d_k$. Our goal is to define $W_K, W_Q$, and a global threshold $\tau$ such that the score $S_{ij} = (\mathbf{x}_i W_Q) \cdot (\mathbf{x}_j W_K)$ exceeds $\tau$ iff $j = \pi(i)$. Our construction works for all vertices $V$ of the graph $G$, independent of the current context in our model; we formalize how this applies to specific contexts below.

**Algorithmic Construction (one-hot case)**  The core idea is to assign each node $v_j$ a random "signature" via its key vector $\mathbf{k}_j$. The query vector $\mathbf{q}_i$ for $v_i$ is the signature of its target, $v_{\pi(i)}$. The dot product between vectors is maximized when the query signature matches the key signature.

---

**Algorithm 4** Construction for Permutation Graphs with One-Hot Inputs

---

1: **Input:** Graph $G = (V, E)$ defined by permutation $\pi$.
2: **Setup:** Choose a probability $p \in (0, 1/2)$, e.g., $p = 1/4$, and dimension $d_k = C \log m$, for sufficiently large constant $C$.
3:
4: **Construct Key Matrix:** Draw $W_K \in \mathbb{R}^{m \times d_k}$ with i.i.d. entries $(W_K)_{jl} \sim \text{Bernoulli}(p)$.
5:    For each node $v_j$, the key is $\mathbf{k}_j = \mathbf{e}_j W_K$.
6:
7: **Construct Query Matrix:** For each node $v_i$, set its query $\mathbf{q}_i = \mathbf{k}_{\pi(i)}$.
8:    This is equivalent to setting the $i$-th row of $W_Q$ to be the $\pi(i)$-th row of $W_K$.
9:
10: **Set Threshold:** $\tau = \frac{p+p^2}{2} d_k$.

---

**Theorem F.1** (Single-head recognition under one-hot inputs). *Under the construction above, for $d_k = C \log m$ with $C$ sufficiently large (depending only on $p$), we have with probability at least $1 - m^{-3}$ over the draw of $W_K$ that*

$$S_{i,\pi(i)} > \tau \quad \text{and} \quad S_{ij} < \tau \text{ for all } i \in V, j \neq \pi(i). \tag{62}$$

*Hence a single attention head correctly identifies all edges of $G$.*

*Proof.* For $j = \pi(i)$,

$$S_{i,\pi(i)} = \mathbf{k}_{\pi(i)} \cdot \mathbf{k}_{\pi(i)} \sim \text{Binomial}(d_k, p) \tag{63}$$

with mean $\mu_1 = d_k p$. For $j \neq \pi(i)$,

$$S_{ij} = \mathbf{k}_{\pi(i)} \cdot \mathbf{k}_j \sim \text{Binomial}(d_k, p^2) \tag{64}$$

with mean $\mu_2 = d_k p^2$. Take $\tau = \frac{\mu_1 + \mu_2}{2} = \frac{p+p^2}{2} d_k$.

For the (lower) tail at the true edge, the Chernoff bound gives

$$\Pr\left[ S_{i,\pi(i)} \leq \tau \right] \leq \exp\left( -\frac{\mu_1 \delta_1^2}{2} \right) \quad \text{where} \quad \delta_1 = 1 - \frac{\tau}{\mu_1} = \frac{1-p}{2}, \tag{65}$$

so $\Pr[S_{i,\pi(i)} \leq \tau] \leq \exp\left( -\frac{d_k p (1-p)^2}{8} \right)$. For the (upper) tail at non-edges, the Chernoff bound yields

$$\Pr\left[ S_{ij} \geq \tau \right] \leq \exp\left( -\frac{\mu_2 \delta_2^2}{2+\delta_2} \right) \quad \text{where} \quad \delta_2 = \frac{\tau}{\mu_2} - 1 = \frac{1-p}{2p}, \tag{66}$$

hence $\Pr[S_{ij} \geq \tau] \leq \exp\left( -\frac{d_k \, p (1-p)^2}{2(1+3p)} \right)$. A union bound over the $m$ target pairs and the $m(m-1)$ non-target pairs gives a total failure probability

$$m \, e^{-c_1 d_k} + m^2 e^{-c_2 d_k} \quad \text{with} \quad c_1 = \frac{p(1-p)^2}{8}, \; c_2 = \frac{p(1-p)^2}{2(1+3p)}. \tag{67}$$

Choosing $d_k = C \log m$ with $C > \max\{3/c_1, 2/c_2\}$ makes this at most $m^{-3}$, establishing the simultaneous separation $S_{i,\pi(i)} > \tau > S_{ij}$ and correctness. $\qquad\square$

Lemma 4.1 immediately now yields correctness for *every* context, independent of context length. Our lower bound from Section D for this case is $\Omega(\frac{m'}{d_{\text{model}}} \log m) = \Omega(\log m)$. Our construction achieves an upper bound of $d_k = O(\log m)$, demonstrating that the bound is tight for this class of problems. Also note that the threshold proof is identical if softmax is used; see Appendix G.

## F.2 CONSTRUCTION II: PERMUTATIONS UNDER COMPRESSIVE EMBEDDINGS

We next prove the correctness of Construction II, which follows from Theorem 4.2, restated here for convenience.

**Theorem F.2** (Multi-head recognition under Gaussian unit-norm embeddings)**.** *Assume the setup and construction of Algorithm 1 with $h = \frac{m}{d_{\text{model}}}$ heads, per-head dimension $d_k = C \log m$ for $C$ sufficiently large, and $\tau = \frac{1}{2} d_k$. If $d_{\text{model}} \geq c_0 \log m$ for a sufficiently large absolute constant $c_0$, then with probability at least $1 - m^{-3}$ over the draw of $(X, W_{\text{sig}})$,*

$$\forall i \in V \ \exists k \in [h] \text{ with } i \in V_k: \quad S^{(k)}_{i, \pi(i)} > \tau \quad and \quad S^{(k)}_{ij} < \tau \ \forall j \neq \pi(i). \tag{68}$$

*Consequently, max-pooling over heads correctly recognizes all edges and $D_K = h \, d_k = \Theta\!\left( \frac{m \log m}{d_{\text{model}}} \right)$.*

*Proof.* Let $\mathbf{u}_i := \mathbf{x}_i X^\top = \mathbf{e}_i(X X^\top)$ and $\boldsymbol{\delta}_i := \mathbf{u}_i - \mathbf{e}_i \in \mathbb{R}^m$. Thus $\mathbf{u}_i$ is the $i$-th row of the Gram matrix $G := X X^\top$; it satisfies $\mathbf{u}_i(i) = 1$ and, for $j \neq i$, $\mathbf{u}_i(j) = \langle \mathbf{x}_i, \mathbf{x}_j \rangle$. Fix a head $k$ and a source $i \in V_k$. As in Construction I, write

$$\mathbf{q}^{(k)}_i = \mathbf{u}_i W'_{Q,(k)}, \qquad \mathbf{k}^{(k)}_j = \mathbf{u}_j W'_{K,(k)}. \tag{69}$$

Using $\mathbf{u}_t = \mathbf{e}_t + \boldsymbol{\delta}_t$ and the definitions of $W'_{Q,(k)}$ and $W'_{K,(k)}$, decompose, for any $j$,

$$\begin{aligned}
S^{(k)}_{ij} &= (\mathbf{u}_i W'_{Q,(k)}) \cdot (\mathbf{u}_j W'_{K,(k)}) \\
&= \underbrace{\mathbf{w}_{\pi(i)} \cdot \mathbf{w}_j \cdot \mathbb{I}(j \in T_k)}_{\text{Signal}} + \underbrace{\mathbf{w}_{\pi(i)} \cdot \sum_{t \in T_k} \delta_{j,t} \mathbf{w}_t}_{N_1} \\
&\quad + \underbrace{\Big( \sum_{s \in V_k} \delta_{i,s} \mathbf{w}_{\pi(s)} \Big) \cdot \mathbf{w}_j \cdot \mathbb{I}(j \in T_k)}_{N_2} + \underbrace{\Big( \sum_{s \in V_k} \delta_{i,s} \mathbf{w}_{\pi(s)} \Big) \cdot \Big( \sum_{t \in T_k} \delta_{j,t} \mathbf{w}_t \Big)}_{N_3}.
\end{aligned}$$

Signal here means the contribution that would remain under a perfect inverse (i.e., if $X X^\top = I$): $\mathbf{w}_{\pi(i)} \cdot \mathbf{w}_j \cdot \mathbb{I}(j \in T_k)$. The Noise terms $N_1, N_2, N_3$ arise solely from the leakage vectors $\boldsymbol{\delta}_i, \boldsymbol{\delta}_j$ due to approximate de-embedding. For $j \in T_k \setminus \{\pi(i)\}$ the cross-inner product $\mathbf{w}_{\pi(i)} \cdot \mathbf{w}_j$ is *not* counted as noise (it is intrinsic signature cross-correlation) and is bounded separately. To bound the Noise terms, we next quantify properties of the approximate inverse $X X^\top$ for unit-norm Gaussian rows.

**Lemma F.3** (Concentration of the approximate inverse)**.** *Let $X$ be as above and $d_{\text{model}} \geq c_0 \log m$ for a sufficiently large constant $c_0$. With probability at least $1 - m^{-4}$, simultaneously for all $i \in [m]$ and heads $k \in [h]$:*

1. $\mathbf{u}_i(i) = 1$ *(deterministically).*

2. *(Leakage $L_2$-mass) For $S \in \{V_k \setminus \{i\}, T_k\}$,*

$$\|\boldsymbol{\delta}_{i,S}\|_2^2 = \sum_{s \in S} \langle \mathbf{x}_i, \mathbf{x}_s \rangle^2 \leq C_2 \tag{70}$$

*for an absolute constant $C_2$ (e.g., $C_2 = 2$).*

3. *(Cross-correlations) For all $i, j$,*

$$\Big| \sum_{a \in T_k} \delta_{i,\pi^{-1}(a)}\, \delta_{j,a} \Big| \;\leq\; C_3 \sqrt{\frac{\log m}{d_{\text{model}}}} \tag{71}$$

*for an absolute constant $C_3$.*

*Proof.* For $j \neq i$, $\langle \mathbf{x}_i, \mathbf{x}_j \rangle$ is mean-zero sub-Gaussian with parameter $\Theta(1/\sqrt{d_{\text{model}}})$, and $\{\langle \mathbf{x}_i, \mathbf{x}_j \rangle\}_{j \in S}$ are independent given $\mathbf{x}_i$. Then $(\langle \mathbf{x}_i, \mathbf{x}_j \rangle^2)_{j \in S}$ are i.i.d. sub-exponential with $\psi_1$-norm $\Theta(1/d_{\text{model}})$ and mean $1/d_{\text{model}}$. For $|S| = d_{\text{model}}$, Bernstein's inequality gives

$$\Pr\Big[ \sum_{s \in S} \langle \mathbf{x}_i, \mathbf{x}_s \rangle^2 > 2 \Big] \;\leq\; e^{-\Omega(d_{\text{model}})}. \tag{72}$$

A union bound over $i$ and the $2h$ choices of $S$ (recall $h = m/d_{\text{model}}$) yields Item 2.

For Item 3, define independent mean-zero sub-exponential variables $Y_a := \langle \mathbf{x}_i, \mathbf{x}_{\pi^{-1}(a)} \rangle \cdot \langle \mathbf{x}_j, \mathbf{x}_a \rangle$ for $a \in T_k$. Each has $\psi_1$-norm $\Theta(1/d_{\text{model}})$ and $\mathbb{E}Y_a = 0$. Bernstein's inequality implies $\Pr\big[\big| \sum_{a \in T_k} Y_a \big| \geq t\big] \leq 2\exp(-\Omega(\min\{d_{\text{model}}t^2,\, d_{\text{model}}t\}))$. Taking $t = C_3\sqrt{(\log m)/d_{\text{model}}}$ and union bounding over all $i, j, k$ proves Item 3 for $c_0$ large enough. Item 1 is immediate from unit-norm rows. $\square$

**Signal.** If $j = \pi(i)$, then Signal $= \|\mathbf{w}_{\pi(i)}\|_2^2 = d_k$ (exactly). If $j \in T_k$ and $j \neq \pi(i)$, then Signal $= \mathbf{w}_{\pi(i)} \cdot \mathbf{w}_j$ is a sum of $d_k$ i.i.d. Rademacher variables and thus sub-Gaussian with mean 0 and variance $d_k$. By a union bound over all $(i, j, k)$, with probability at least $1 - m^{-5}$,

$$|\text{Signal}| \;\leq\; C_\star \sqrt{d_k \log m} \qquad \text{for all } (i, j \in T_k \setminus \{\pi(i)\}, k), \tag{73}$$

for an absolute constant $C_\star$.

**Noise.** Condition on $X$ and apply Lemma F.3. For $N_1$,

$$N_1 = \sum_{r=1}^{d_k} \Big( \sum_{t \in T_k} \delta_{j,t}\, w_t[r] \Big) w_{\pi(i)}[r] \tag{74}$$

is a sum of $d_k$ i.i.d. mean-zero sub-Gaussian variables with variance proxy $\|\boldsymbol{\delta}_{j,T_k}\|_2^2 \leq C_2$. Hence, by Bernstein/Hoeffding and a union bound over $(i, j, k)$,

$$|N_1| \;\leq\; C_4 \sqrt{C_2\, d_k\, \log m} \tag{75}$$

holds w.h.p. for an absolute constant $C_4$. The same bound holds for $N_2$ with $\|\boldsymbol{\delta}_{i,V_k}\|_2^2 \leq C_2$.

For $N_3$, write for each column $r$,

$$X_r := \sum_{s \in V_k} \delta_{i,s}\, w_{\pi(s)}[r], \qquad Y_r := \sum_{t \in T_k} \delta_{j,t}\, w_t[r]. \tag{76}$$

Then $N_3 = \sum_{r=1}^{d_k} X_r Y_r$. Conditional on $X$, $\{(X_r, Y_r)\}_{r=1}^{d_k}$ are i.i.d.; each $X_r$ and $Y_r$ is mean-zero sub-Gaussian with parameters $\lesssim \|\boldsymbol{\delta}_{i,V_k}\|_2 \leq \sqrt{C_2}$ and $\lesssim \|\boldsymbol{\delta}_{j,T_k}\|_2 \leq \sqrt{C_2}$, respectively. Thus $X_r Y_r$ is mean $\langle \boldsymbol{\delta}_{i,\pi^{-1}(T_k)}, \boldsymbol{\delta}_{j,T_k} \rangle$ and sub-exponential with $\psi_1$-norm $\lesssim C_2$. Consequently,

$$\mathbb{E}[N_3 \mid X] \;=\; d_k \big\langle \boldsymbol{\delta}_{i,\pi^{-1}(T_k)}, \boldsymbol{\delta}_{j,T_k} \big\rangle, \tag{77}$$

and, by Bernstein plus a union bound,

$$\big| N_3 - \mathbb{E}[N_3 \mid X] \big| \;\leq\; C_5\, C_2\, \sqrt{d_k \log m} \tag{78}$$

w.h.p. for an absolute constant $C_5$. Using Lemma F.3(3),

$$\big| \mathbb{E}[N_3 \mid X] \big| \;\leq\; d_k\, C_3 \sqrt{\frac{\log m}{d_{\text{model}}}}. \tag{79}$$

**Separation.** Choose constants $c_0, C$ large enough so that

$$C_3 \sqrt{\frac{\log m}{d_{\mathrm{model}}}} \ \leq \ \frac{1}{16} \qquad \text{and} \qquad \left(C_\star + 2 C_4 \sqrt{C_2} + C_5 C_2\right)\sqrt{\frac{\log m}{d_k}} \ \leq \ \frac{1}{16}. \tag{80}$$

This is feasible since $d_{\mathrm{model}} \geq c_0 \log m$ and $d_k = C \log m$.

*Target edge $j = \pi(i)$.* Using the bounds above (recall Signal $= d_k$ exactly),

$$S_{i,\pi(i)}^{(k)} \ \geq \ d_k - \underbrace{\left(C_4 \sqrt{C_2\, d_k \log m}\right)}_{|N_1|} - \underbrace{\left(C_4 \sqrt{C_2\, d_k \log m}\right)}_{|N_2|} - \underbrace{\left(C_5 C_2 \sqrt{d_k \log m} + \tfrac{1}{16} d_k\right)}_{|N_3|} \ > \ \tfrac{3}{4} d_k \ > \ \tau. $$
$$\tag{81}$$

*Non-edge $j \neq \pi(i)$.* If $j \notin T_k$ then Signal $= 0$ and $N_2 = 0$, so

$$|S_{ij}^{(k)}| \ \leq \ C_4 \sqrt{C_2\, d_k \log m} + \left(C_5 C_2 \sqrt{d_k \log m} + \tfrac{1}{16} d_k\right) \ < \ \tfrac{1}{4} d_k \ < \ \tau. \tag{82}$$

If $j \in T_k \setminus \{\pi(i)\}$, then $|\text{Signal}| \leq C_\star \sqrt{d_k \log m}$ and the same bounds for $N_1, N_2, N_3$ apply, giving $|S_{ij}^{(k)}| < \tfrac{1}{4} d_k < \tau$.

A union bound over all $(i, j, k)$ completes the proof. $\qquad\square$

Thus, with Gaussian unit-norm embeddings and Rademacher signatures, our construction recognizes the entire graph using a total key dimension

$$D_K = h \cdot d_k = O\!\left(\frac{m \log m}{d_{\mathrm{model}}}\right), \tag{83}$$

This bound is asymptotically optimal, matching our lower bound within a constant factor. In the proof of Theorem 4.2, the non-edge bounds hold uniformly over all $(i, j, k)$ (we union bound over $(i, j, k)$), so for any $j \neq \pi(i)$ we have $S_{ij}^{(k)} < \tau$ for *all* heads $k$ and hence $S_{ij}^{\max} < \tau$, while the target edge satisfies $S_{i,\pi(i)}^{\max} > \tau$. Lemma 4.1 then yields correctness on $E|_\mathcal{C}$ for every context $\mathcal{C}$.

### F.3 CONSTRUCTION III: MORE GENERAL EMBEDDINGS

The analysis of Construction II (Gaussian unit–norm) ultimately used only two facts about the Gram matrix $XX^\top$: (i) diagonals concentrate around a common scale, and (ii) for any *small* subset of indices the off–diagonal leakage has bounded $\ell_2$ mass, with a mild control on a corresponding cross–leakage term. We package these into a reusable, block–level notion that subsumes the usual pairwise incoherence and is tight enough to cover sparse/binary compressive embeddings.

**Definition F.4** (Restricted self–incoherence at block size $B$). Fix parameters $\mu > 0$, $\varepsilon_d \in [0, 1)$, block size $B \in \mathbb{N}$, and leakage levels $\rho, \gamma \geq 0$. An embedding matrix $X \in \mathbb{R}^{m \times d_{\mathrm{model}}}$ with rows $\{\mathbf{x}_i\}_{i=1}^m$ is $(\mu, \varepsilon_d, B; \rho, \gamma)$–*restricted self–incoherent* if, writing

$$X_{\mathrm{inv}} := \frac{1}{\mu} X^\top, \qquad \mathbf{u}_i := \mathbf{x}_i X_{\mathrm{inv}} = \frac{1}{\mu} \mathbf{e}_i (XX^\top), \qquad \boldsymbol{\delta}_i := \mathbf{u}_i - \mathbf{e}_i, \tag{84}$$

the following hold simultaneously:

1. **Diagonal stability:** $\mathbf{u}_i(i) \in [1 - \varepsilon_d, \, 1 + \varepsilon_d]$ for all $i$.

2. **Restricted leakage mass:** for every $i$ and every $S \subseteq [m] \setminus \{i\}$ with $|S| \leq B$,

$$\|\boldsymbol{\delta}_{i,S}\|_2^2 = \sum_{s \in S} \delta_i(s)^2 \ \leq \ \rho. \tag{85}$$

3. **Restricted cross–leakage:** for every $i, j$ and $S \subseteq [m]$ with $|S| \leq B$,

$$\Big| \sum_{a \in S} \delta_i(a)\, \delta_j(a) \Big| \ \leq \ \gamma. \tag{86}$$

---

**Algorithm 5** Construction for Generalized Embeddings

---

1: **Input:** Embedding matrix $X \in \mathbb{R}^{m \times d_{\text{model}}}$; permutation graph $G = (V, E)$ with $\pi : V \to V$.
2: **Parameters:** Signature sparsity $p \in (0, 1/20]$; per–head width $d_k = C \log m$ for a sufficiently large absolute constant $C$.
3: **Random signatures:** Draw $W_{\text{sig}} \in \{0, 1\}^{m \times d_k}$ with i.i.d. Bernoulli$(p)$ entries; let $\mathbf{w}_j$ denote its $j$-th row.
4: **Set Threshold:** $\tau := \frac{p+p^2}{2}\, d_k$.
5: **Choose block size and partition:** Pick a block size $B$ (specified per embedding family below). Let $h := \lceil m/B \rceil$ and partition $V$ into blocks $V_1, \ldots, V_h$ with $|V_k| \leq B$. For each head $k$, define its target set

$$T_k \ := \ \{\pi(s) : s \in V_k\}. \tag{87}$$

6: **Define one-hot–space templates (for each head $k$):**

$$W'_{Q,(k)}(i,:) \ = \ \begin{cases} \mathbf{w}_{\pi(i)} & i \in V_k \\ 0 & \text{else} \end{cases}, \qquad W'_{K,(k)}(j,:) \ = \ \begin{cases} \mathbf{w}_j & j \in T_k \\ 0 & \text{else} \end{cases}. \tag{88}$$

7: **Realize parameters via approximate inverse:**

$$W_Q^{(k)} \ = \ X_{\text{inv}}\, W'_{Q,(k)}, \qquad W_K^{(k)} \ = \ X_{\text{inv}}\, W'_{K,(k)}. \tag{89}$$

---

**Theorem F.5** (Recognition under restricted self–incoherence). *Let $X$ be $(\mu, \varepsilon_d, B; \rho, \gamma)$–restricted self–incoherent for some $B$. Fix any $p \leq 1/20$, take $d_k = C \log m$ with $C$ a sufficiently large absolute constant, and set $\tau = \frac{p+p^2}{2} d_k$. There exist absolute numerical constants $(c_1, c_2, c_3)$ such that if*

$$\varepsilon_d \leq c_1, \qquad \rho \ \leq \ \frac{c_2}{\log m}, \qquad \gamma \ \leq \ \frac{c_3}{\log m}, \tag{90}$$

*then with probability at least $1 - m^{-3}$ over $W_{\text{sig}}$ (and the draw of $X$ if random),*

$$\forall i \in V \ \exists k \in [h] \ with \ i \in V_k: \quad S_{i,\pi(i)}^{(k)} > \tau \quad and \quad S_{ij}^{(k)} < \tau \ \ \forall j \neq \pi(i). \tag{91}$$

*Consequently, max–pooling over heads recovers all edges and the total key budget satisfies*

$$D_K = h\, d_k = \Theta\Big(\frac{m \log m}{B}\Big). \tag{92}$$

*Proof sketch.* As in Construction II, the score decomposes into a *signal* term plus three *noise* terms: $S_{ij}^{(k)} = \mathbf{w}_{\pi(i)} \cdot \mathbf{w}_j \cdot \mathbb{I}(j \in T_k) + N_1 + N_2 + N_3$, with $N_1, N_2, N_3$ arising from $\boldsymbol{\delta}_i, \boldsymbol{\delta}_j$. Write $\mathbf{u}_t = \mathbf{e}_t + \boldsymbol{\delta}_t$ and expand $S_{ij}^{(k)}$ as in Construction II. Conditioned on $X$, each column of $W_{\text{sig}}$ contributes an independent copy of the signal/noise decomposition. Using Chernoff for the Bernoulli signal coordinates gives, uniformly over all $(i, j, k)$, the standard separation $\mu_1 - \mu_2 = (p - p^2)d_k$ between $j = \pi(i)$ and $j \in T_k \setminus \{\pi(i)\}$ up to $O(\sqrt{d_k \log m})$ fluctuations.

For $N_1$ and $N_2$, restricted leakage mass yields $\text{Var}(N_1), \text{Var}(N_2) \ \lesssim \ d_k \rho$ and hence $|N_1|, |N_2| \ \lesssim \ \sqrt{d_k \rho \log m}$ uniformly with probability $1 - m^{-5}$. For $N_3$, the centered part concentrates at scale

$\lesssim \sqrt{d_k \log m} \cdot (\rho)^{1/2}$, while the mean shift equals $d_k \langle \boldsymbol{\delta}_{i,\pi^{-1}(T_k)}, \boldsymbol{\delta}_{j,T_k} \rangle$ and is controlled by $\gamma$. Choosing $C$ large and $(c_1, c_2, c_3)$ small makes the total noise $< \frac{1}{4}(p - p^2)d_k$ uniformly, while the target signal sits $> \frac{3}{4}(p - p^2)d_k$ above $\mu_2$, giving the stated threshold separation. $\qquad \square$

**How to pick $B$.** The theorem asks only that $\rho, \gamma \lesssim 1/\log m$ at the chosen block size $B$. Different embedding families admit different $(\rho, \gamma)$–vs–$B$ trade–offs; plugging the corresponding $B$ into $D_K = \Theta((m/B)\log m)$ yields the budget.

COROLLARIES FOR COMMON EMBEDDING MODELS

**Corollary F.6** (Gaussian unit–norm (GUN)). *Let each row $\mathbf{x}_i$ be drawn i.i.d. as $\tilde{\mathbf{x}}_i \sim \mathcal{N}(0, I/d_{model})$ and then $\ell_2$–normalized. Then w.h.p.*

$$\varepsilon_d = 0, \qquad \rho \lesssim \frac{B}{d_{model}}, \qquad \gamma \lesssim \sqrt{\frac{B}{d_{model}}}, \tag{93}$$

*and Theorem F.5 holds for any $B \le c\, d_{model}/\log m$. Choosing $B = \Theta(d_{model})$ yields*

$$h = \Theta\Big(\frac{m}{d_{model}}\Big), \qquad D_K = \Theta\Big(\frac{m \log m}{d_{model}}\Big), \tag{94}$$

*in agreement with Theorem 4.2 up to constants (the specialized proof in Construction II attains this with the sharp choice $B = d_{model}$).*

**Corollary F.7** (Random binary compressive embeddings (RBCE)). *Let $X \in \{0,1\}^{m \times d_{model}}$ have i.i.d. Bernoulli$(p_B)$ entries with $p_B = \Theta(\log m/d_{model})$ (sparse binary features). Set $\mu := d_{model}p_B$. Then with probability at least $1 - m^{-4}$ the following hold simultaneously:*

$$\mathbf{u}_i(i) \in [1 - \varepsilon_d, 1 + \varepsilon_d] \text{ with } \varepsilon_d \lesssim \frac{1}{\sqrt{\mu}}, \qquad \rho \lesssim \frac{B}{d_{model}}, \qquad \gamma \lesssim B\, p_B^2. \tag{95}$$

*Consequently, taking*

$$B = \Theta\Big(\frac{d_{model}}{\log m}\Big) \implies \rho \lesssim \frac{1}{\log m}, \ \gamma \lesssim \frac{\log m}{d_{model}}, \tag{96}$$

*and Theorem F.5 applies. The number of heads and total key budget become*

$$h = \Theta\Big(\frac{m \log m}{d_{model}}\Big), \qquad D_K = h\, d_k = \Theta\Big(\frac{m \log^2 m}{d_{model}}\Big). \tag{97}$$

*Proof idea for Corollary F.7.* Row norms are Binomial$(d_{model}, p_B)$ and concentrate at $\mu$ with relative error $O(1/\sqrt{\mu})$ by Chernoff, giving the $\varepsilon_d$ bound. For a fixed $i$ and any $S$ with $|S| \le B$,

$$\sum_{s \in S} \langle \mathbf{x}_i, \mathbf{x}_s \rangle^2 \le \sum_{s \in S} \langle \mathbf{x}_i, \mathbf{x}_s \rangle \quad \text{and} \quad \mathbb{E}\big[\langle \mathbf{x}_i, \mathbf{x}_s \rangle\big] = d_{model}p_B^2, \tag{98}$$

so $\mathbb{E}\|\boldsymbol{\delta}_{i,S}\|_2^2 = \frac{1}{\mu^2} \sum_{s \in S} \mathbb{E}\langle \mathbf{x}_i, \mathbf{x}_s \rangle^2 \lesssim B/d_{model}$, and a Bernstein + union bound yields $\rho \lesssim B/d_{model}$. Similarly, $\mathbb{E} \sum_{a \in S} \delta_i(a)\delta_j(a) = \frac{|S|}{\mu^2} \mathbb{E}\langle \mathbf{x}_i, \mathbf{x}_a \rangle \mathbb{E}\langle \mathbf{x}_j, \mathbf{x}_a \rangle \lesssim Bp_B^2$, and concentration gives $\gamma \lesssim Bp_B^2$ uniformly. $\qquad \square$

**Signature family.** We stated the construction with Bernoulli$(p)$ signatures because the thresholding analysis naturally separates $j = \pi(i)$ from $j \in T_k \setminus \{\pi(i)\}$ at means $pd_k$ vs. $p^2 d_k$. One can equivalently use Rademacher $\{\pm 1\}$ signatures with threshold $\tau = \frac{1}{2}d_k$; all bounds above translate verbatim with the same $B$ and $d_k = \Theta(\log m)$.

**Takeaways.** Definition F.4 abstracts the only geometric inputs needed by the attention construction. Plugging in model–specific $(\rho, \gamma)$–vs–$B$ trade–offs yields the head count $h = \Theta(m/B)$ and total key budget $D_K = \Theta((m/B)\log m)$. For Gaussian unit–norm embeddings one recovers the $D_K = \Theta(m \log m/d_{\text{model}})$ guarantee; for sparse random binary compressive embeddings one obtains $D_K = \Theta(m \log^2 m/d_{\text{model}})$.

### F.4 Construction IV: General Graphs

We now extend the permutation constructions to general directed graphs $G = (V, E)$ with $|V| = m$ vertices and $|E| = m'$ edges. In this case, our information theoretic lower bound on total key dimension is $D_K = \Omega\left(\frac{m'}{d_{\text{model}}} \log(m^2/m')\right)$. We here provide a general upper bound for any graph, and show that for graphs that have a mild skew condition (the maximum degree is not too much larger than the average degree), it asymptotically matches this lower bound for all but the densest graphs (which match within a log factor). As before we use max aggregation over heads with a global scalar threshold $\tau$, and we work under the *Gaussian unit-norm embedding* model from Construction II: the row vectors of $X \in \mathbb{R}^{m \times d_{\text{model}}}$ are i.i.d. isotropic Gaussian followed by $L_2$-normalization. All probabilities are over the draw of $X$ and of the (head-shared) random signature matrix.

**Packing edges into matchings.** The analysis in Theorem 4.2 operates on blocks in which each source has exactly one outgoing edge *and* targets are distinct within the block. Equivalently, each head should see a *matching* (a partial permutation) between a set of sources and a set of targets.

We will use a simple decompositions of the edge set into matchings of size $d_{\text{model}}$ which will be our block size. Write $d_{\text{out}}(i)$ and $d_{\text{in}}(i)$ for the out-/in-degree of $v_i$. Let $\Delta_{\text{out}} := \max_i d_{\text{out}}(i)$ and $\Delta_{\text{in}} := \max_i d_{\text{in}}(i)$ denote the maximum out- and in-degrees, and write $\Delta := \max\{\Delta_{\text{out}}, \Delta_{\text{in}}\}$.

**Lemma F.8** (Coloring-and-batching decomposition). *Let $G = (V, E)$ be any directed graph on $m$ vertices and $m'$ edges, and let $H := \left\lceil \frac{m'}{d_{model}} \right\rceil + \Delta$. Then there exists a partition of $E$ into $H$ disjoint sets $M_1, \ldots, M_H$ such that for every $k$: (i) $M_k$ is a matching (no two edges in $M_k$ share a source or a target); (ii) $|M_k| \leq d_{model}$.*

*Proof.* Identify $G$ with its bipartite incidence graph $\mathcal{B} = (V_L \cup V_R, E)$ where each directed edge $(i, j)$ becomes an undirected edge between $i \in V_L$ and $j \in V_R$. Then $\Delta(\mathcal{B}) = \Delta$. By Kőnig's line-coloring theorem, $E = F_1 \cup \cdots \cup F_\Delta$ with each $F_c$ a matching. Split each $F_c$ into blocks of size at most $d_{\text{model}}$; since $\sum_{c=1}^{\Delta} \lceil |F_c|/d_{\text{model}} \rceil \leq \left\lceil \frac{\sum_c |F_c|}{d_{\text{model}}} \right\rceil + \Delta = \left\lceil \frac{m'}{d_{\text{model}}} \right\rceil + \Delta = H$, we obtain $H$ matchings $M_k$ each of size at most $d_{\text{model}}$. $\square$

Thus, after packing via Lemma F.8 head $k$ will operate on the matching $M_k$. Let $V_k \subseteq V$ and $T_k \subseteq V$ denote the sources and targets incident to $M_k$ and write $\pi_k : V_k \to T_k$ for the bijection defined by $M_k$.

**Construction.** We reuse the compressive permutation machinery head-by-head.

**Theorem F.9** (General graphs). *Assume $d_{model} \geq c_0 \log m$ for a sufficiently large constant $c_0$. With the construction above (using $h = \left\lceil \frac{m'}{d_{model}} \right\rceil + \Delta$ heads and $d_k = C \log m$), there is a universal $C$ such that, with probability at least $1 - m^{-3}$ over the draw of $(X, W_{\text{sig}})$, simultaneously for all ordered pairs $(i, j)$,*

$$S_{ij}^{\max} = \max_{1 \leq k \leq H} S_{ij}^{(k)} \begin{cases} > \tau & \text{if } (i,j) \in E, \\ < \tau & \text{if } (i,j) \notin E. \end{cases} \tag{101}$$

*Consequently, $D_K = O\left(\frac{m' \log m}{d_{model}} + \Delta \log m\right)$.*

---

**Algorithm 6** Construction for General Graphs

---

1: **Input:** Directed graph $G = (V, E)$ with $|V| = m$, $|E| = m'$; embedding matrix $X \in \mathbb{R}^{m \times d_{\text{model}}}$ with Gaussian unit-norm rows.

2: **Parameters:** Number of heads $h = H = \left\lceil \frac{m'}{d_{\text{model}}} \right\rceil + \Delta$; per-head key/query dimension $d_k = C \log m$ for a sufficiently large absolute constant $C$. Each head uses block size $d_{\text{model}}$.

3: **Pack edges into matchings:** Decompose $E$ into disjoint matchings $M_1, \ldots, M_H$ with $|M_k| \leq d_{\text{model}}$ using Lemma F.8. For each $k$, let $V_k$ and $T_k$ be the sources and targets incident to $M_k$ and write $\pi_k : V_k \to T_k$ for the associated bijection.

4: **Random signatures:** Draw a shared Rademacher matrix $W_{\text{sig}} \in \{\pm 1\}^{m \times d_k}$ with i.i.d. entries; let $\mathbf{w}_j$ denote its $j$-th row.

5: **Per-head "ideal" matrices:**

$$\left(W'_{Q,(k)}\right)_{i,\cdot} := \begin{cases} \mathbf{w}_{\pi_k(i)} & i \in V_k \\ \mathbf{0} & \text{otherwise} \end{cases}, \qquad \left(W'_{K,(k)}\right)_{j,\cdot} := \begin{cases} \mathbf{w}_j & j \in T_k \\ \mathbf{0} & \text{otherwise} \end{cases}, \qquad (99)$$

where $\mathbf{w}_j$ is the $j$-th row of $W_{\text{sig}}$.

6: **Final projections (approximate de-embedding):** As in Construction II, use the approximate inverse $X^\top$:

$$W_Q^{(k)} = X^\top W'_{Q,(k)}, \qquad W_K^{(k)} = X^\top W'_{K,(k)}. \qquad (100)$$

7: **Set Threshold:** $\tau = \frac{1}{2} d_k$.

---

*Proof sketch.* By Lemma F.8, each head $k$ sees a matching $M_k$ of size at most $d_{\text{model}}$, with a bijection $\pi_k : V_k \to T_k$. Within head $k$, the score decomposition and concentration bounds are exactly those of Theorem 4.2: for $(i, j) = (i, \pi_k(i))$ the Signal term equals $d_k$ and the three Noise terms $(N_1, N_2, N_3)$ are controlled using Lemma F.3, since all leakage sets ($V_k \setminus \{i\}$ and $T_k$) have size $\leq d_{\text{model}}$. For $(i, j) \neq (i, \pi_k(i))$, Signal is a sum of i.i.d. Rademachers with variance $d_k$, while the same leakage bounds control $N_1, N_2, N_3$. Choosing $C$ and $c_0$ as in Theorem 4.2 yields, within each head, $S^{(k)}_{i,\pi_k(i)} > \tau$ and $|S^{(k)}_{ij}| < \tau$ for all $j \neq \pi_k(i)$ simultaneously with probability $1 - m^{-4}$.

A union bound over all heads and all pairs in those heads costs only a $\log$ factor absorbed by $d_k = C \log m$: using $|M_k| \leq d_{\text{model}}$ and $\sum_k |M_k| = m'$, we have $\sum_k |M_k|^2 \leq d_{\text{model}} \sum_k |M_k| = d_{\text{model}} m' = O(m' d_{\text{model}})$ events in total. Finally, max pooling across heads preserves separation (non-edges are below $\tau$ *in every head*, and each true edge belongs to exactly one $M_k$), and Lemma 4.1 yields context-robustness for arbitrary subsets $\mathcal{C} \subseteq V$. □

**Degree skew and tightness.** Let $d_{\text{avg}} = \frac{m'}{m}$. Define the *skew factor* to be $\Delta / d_{\text{avg}}$ and consider the condition

$$\frac{\Delta}{d_{\text{avg}}} \leq \frac{m}{d_{\text{model}}}. \qquad (102)$$

In other words, the ratio of the maximum degree to the average degree is no larger than the compression of the embedding, or equivalently, $\Delta \leq \frac{m'}{d_{\text{model}}}$. This condition automatically holds for all $d$-regular graphs (since $\Delta = d_{\text{avg}}$).

**Corollary F.10** (Bounded Skew). *Assume $d_{model} \geq c_0 \log m$ and (102). Then the construction with $h_0 = \lceil m'/d_{model} \rceil$ heads and $d_k = C \log m$ achieves the same separation guarantee as Theorem F.9, and $D_K = \Theta\left(\frac{m' \log m'}{d_{model}}\right)$.*

This is immediate from from Theorem F.9 and asymptotically matches the lower bound from Section D for this class of graphs, provided $m' = O(m^{2-\epsilon})$ for some positive constant $\epsilon$.

# G   ADDITIONAL JUSTIFICATION FOR THE MODEL (SECTION 3)

**Computational footprint.**   While it is natural to consider the number of heads ($h$) and the per-head key/query dimension ($d_k$) as two separate resources, we argue that the most relevant complexity measure is their product. In practice, the computation for multiple heads is not performed as $h$ distinct operations but as a single, larger batched operation. Let $\mathbf{X} \in \mathbb{R}^{\ell \times d_{\text{model}}}$ stack the context embeddings. With $W_Q^{\text{cat}}, W_K^{\text{cat}} \in \mathbb{R}^{d_{\text{model}} \times (h d_k)}$ formed by concatenating head weights, queries/keys are $\mathbf{Q}_{\text{total}} = \mathbf{X} W_Q^{\text{cat}}$ and $\mathbf{K}_{\text{total}} = \mathbf{X} W_K^{\text{cat}}$. Both flops and parameter/memory cost scale as $O(\ell\, d_{\text{model}}\, h d_k)$ and $O(d_{\text{model}}\, h d_k)$, respectively, motivating $D_K = h d_k$ as the budget.

While sub-cubic matrix multiplication algorithms could theoretically make one large head asymptotically faster than several smaller ones, this effect is absent in practice. The true bottleneck for these operations on modern hardware is *memory bandwidth*—the rate at which the matrices can be fetched from memory. The total data moved is proportional to the size of the weight matrices, which is in turn proportional to $d_{\text{model}} \cdot (h \cdot d_k)$. Because deep learning libraries are highly optimized to perform these batched multiplications, the performance typically tracks the total size of the key and query matrices, regardless of the number of heads.

**Analyzing the QK channel in isolation.**   Since RGR asks *where* a source should connect, the OV pathway only reweights or propagates information after the routing decision has been made. Thus, a correct edge can only dominate if the QK gate already concentrates sufficient mass on the true target. Increasing value dimension $D_V$ amplifies whatever QK selects; it does not fix mis-routing.[5] Furthermore, our construction aligns with recent mechanistic analyses that separate each attention head into an OV circuit (what is read/written) and a QK circuit (where to attend); the QK circuit determines the attention pattern and thus the directed edges in RGR (32; 18). To further justify this abstraction, we study the impact of incorporating the value channel in Appendices B and C.3.

**Aggregating across heads.**   This 'OR-of-heads' max is an analysis device for the edge test; it does not assert cross-head QK interaction in standard MHA implementations (where heads are combined in the value pathway). However, aggregating multi-head QK scores by a max implements the intended RGR semantics: each head $k$ specializes to a relational template, and an (untyped) edge should exist if *any* template fires, i.e., $S_{pq}^{\max} = \max_k S_{pq}^{(k)} > \tau$ (an OR-of-relations). If a smooth alternative is desired, replacing max with log-sum-exp preserves the binary decision up to a global threshold shift, since for any scores $a_1, \ldots, a_h$,

$$\max_k a_k \;\le\; \mathrm{LSE}(a) = \log \sum_{k=1}^{h} e^{a_k} \;\le\; \max_k a_k + \log h, \tag{103}$$

so one can retune $\tau$ by at most $\log h$ without changing the classifier (6).

However, we do also consider a more standard aggregation model in Appendices A and C.2. There, we consider softmax over items in the context with the usual scaling inside the logits. Initially, as in our base model, we isolate the QK channel and omit values. Without a value channel, any permutation-invariant linear readout across heads reduces (up to scaling) to summing softmax masses, so we aggregate heads via summation. We then augment this model with a value channel in Appendix B.

---

[5]Note that our model is scoped to a single self-attention layer; multi-layer iterative routing is outside our abstraction.

## H   RELATED WORK

We survey work most relevant to our capacity-centric view of self-attention and position our **Relational Graph Recognition** (RGR) results in that landscape. The central distinction we draw is between *what* attention can compute in principle (expressivity), *how* architectural resources govern this power (capacity), and *which* parts of the Transformer carry the binding/addressing load (keys/queries vs. other channels).

### MEMORIZATION CAPACITY AND PARAMETER–DEPENDENT BOUNDS

A growing body of work quantifies how many input–label associations Transformers—and, more narrowly, the attention mechanism—can memorize. As described in Section 2, the bounds from the memorization setting do not directly imply bounds on RGR, nor the other way around. For the attention module itself, (41) prove that a single MHA layer with $h$ heads can memorize $\Omega\big(h \min\{\ell, d_k\}\big)$ examples under a linear-independence assumption on the inputs, highlighting linear scaling in $h$ and the role of the per-head key/query width $d_k$. Complementary analyses bound attention's memory depth and clarify depth–capacity trade-offs (40). Moving to full Transformers, constructive results show that (under token-wise $(r,\delta)$-separated inputs) a stack of $2\ell$ self-attention layers suffices to memorize $N$ sequences with $\tilde{O}(\ell + \sqrt{\ell N})$ parameters (34); even a *single-layer, single-head* Transformer has nontrivial capacity under the same separatedness assumption, whereas replacing softmax by hardmax breaks memorization (30).

Beyond construction-style bounds, (40) give general upper and lower bounds for next-token prediction that scale as $\Theta(\omega N)$ in the presence of positional encodings and a vocabulary of size $\omega$, and (8) show that a *single-layer* Transformer can memorize when sequences are sufficiently zero-padded (though not in a parameter-optimal way). Classical results for ReLU networks connect parameter counts to memorization thresholds and VC-style capacity (55; 56). More closely related to our focus on resource efficiency, (31) establish nearly matching upper/lower bounds on the *minimal parameter count* needed for memorization in Transformers: $\tilde{O}(\sqrt{N})$ parameters are sufficient (and necessary up to logs) for next-token prediction, and $\tilde{O}(\sqrt{\ell N})$ for sequence-to-sequence, under token-wise separatedness; they further suggest that self-attention effectively *identifies* sequences while the feed-forward network can become the bottleneck when *associating* labels.

### SUPERPOSITION, CONSTRUCTIVE DESIGNS, AND DEPTH SEPARATION

A concurrent line of work analyzes how networks compute many features in *superposition*, with lower and upper bounds for narrow MLPs and constructive designs for multi-feature computation (2; 1; 25). The capacity limits shown there (stated as upper and lower bounds on neurons to compute a number of Boolean functions in parallel) are complementary to ours in terms of architectures: their focus is on MLPs while ours is on attention.

Foundational depth-separation and minimal-width universality results motivate the proof template we adopt—information-theoretic lower bounds matched by explicit constructions (52; 24; 33; 11). In attention, constructive correspondences also explain how multi-head architectures partition pattern spaces; e.g., with relative positions, $s^2$ heads can realize any $s \times s$ convolution (10). Our constructions similarly partition relational signal across heads to mitigate interference when $d_{\text{model}} \ll m$, explaining the empirical advantage of many small heads and clarifying when too-small $d_k$ triggers low-rank failure (5).

### DIMENSION-, RANK-, AND RESOURCE–DRIVEN EXPRESSIVITY

A growing body of theory isolates how *dimensional* resources govern attention's representational power. Universality guarantees establish that sufficiently resourced Transformers can approximate sequence-to-

sequence functions (67), while more refined results show task-dependent strengths and weaknesses (49). Focusing on the attention map, (38) prove that with fixed error and sparsity, self-attention can approximate dynamic sparse right-stochastic matrices using only $O(\log \ell)$ hidden dimensions (for context length $\ell$), echoing the role of near-orthogonality we exploit in our constructions. Conversely, (5) identify a per-head *low-rank bottleneck*: when $d_k < \ell$, a head cannot realize arbitrary $\ell \times \ell$ stochastic attention matrices. This clarifies a trade-off inside the total key/query budget $D_K = h \, d_k$: pushing $D_K$ into many tiny heads can induce head-wise rank limits.

Beyond these, several works develop structural and inductive-bias characterizations of self-attention. (14) show that *pure* attention without mixing loses rank doubly-exponentially with depth, explaining failure modes in deep attention stacks and underscoring the role of residual mixing. (16) analyze *variable creation* and sparsity patterns induced by softmax, while (48) use convex duality to give optimization- and geometry-based interpretations of ViT attention. For sample complexity and approximation, (36) study learning and generalization of shallow ViTs; rates and approximation guarantees have been developed for Transformer encoders and sequence models (22; 51; 29). Recent generalization bounds that are (largely) sequence-length independent sharpen this picture (53). Finally, theory has also pinpointed sparsity-oriented inductive biases: Transformers provably learn *sparse token selection* that FCNs cannot (65), and exhibit a simplicity bias for sparse Boolean functions (4).

Empirical observations likewise single out the key/query channel as an operative budget. Our results formalize this perspective for a concrete family (RGR), deriving matching lower and constructive upper bounds in terms of $D_K$ and the number of relations.

FORMAL-LANGUAGE LIMITS, COMPOSITIONALITY, AND UNIVERSALITY

Formal-language analyses delimit what fixed-size attention can recognize. Beyond general universality (67), there are sharp impossibility results for periodic and hierarchical languages (23; 3; 66). Recent work uses communication-complexity arguments to show single-layer self-attention struggles with *function composition* at fixed embedding/heads, e.g., "grandparent-of" requires resources that scale with domain size (45). Complementing these, (39) identify additional structural constraints on what Transformers can compute under realistic resource regimes. We view these results as orthogonal to RGR: they characterize *classes of computations*, whereas we fix a relational family and ask *how much key/query budget* is necessary and sufficient to represent its edges across arbitrary contexts.

IN-CONTEXT LEARNING AND ALGORITHMIC VIEWS OF SELF-ATTENTION

A complementary line of theory frames Transformers—and attention in particular—as executing *algorithms* over the context. (37) analyze generalization and implicit model selection in in-context learning; (62) give evidence that Transformers can implement gradient-descent-like updates in context; and (19) characterize which simple function classes are learnable in context. These works clarify how attention can implement algorithmic behaviors over token sets, while our RGR focus quantifies the *key–query capacity* required to retrieve relational edges reliably.

CONNECTIVITY PATTERNS VS. CAPACITY IN THE KEY/QUERY CHANNEL

An alternative way to constrain attention is by controlling the connectivity pattern of the attention graph. Even $O(\ell)$-sparse patterns can be universal under appropriate designs (68), and systematic pruning of dense patterns maps out cost–performance frontiers (64). Our analysis treats connectivity as *not* the bottleneck: given the ability to attend broadly, the limiting factor for RGR is how much relational information can be encoded and separated in keys/queries as $m$ and $m'$ grow.

## HEAD SPECIALIZATION, PRUNING, AND INFORMATION BOTTLENECKS

Mechanistic interpretability consistently finds that specific heads specialize to linguistic relations (9; 60). At the same time, many trained heads can be pruned with small accuracy loss (43; 61), indicating redundancy. Information-bottleneck analyses at the head/layer level quantify such redundancy and attribution in both language and vision models (46; 26), and architectural proposals target representation bottlenecks (20). Our results supply a capacity-theoretic backbone for these observations: for RGR, performance transitions are governed primarily by $D_K$; distributing $D_K$ across heads reduces interference between superposed relations, but overly small $d_k$ per head incurs rank limits—predicting both specialization and safe pruning regimes.

## ATTENTION AS ASSOCIATIVE MEMORY VS. RELATIONAL ADDRESSING

Modern Hopfield networks are equivalent, in a precise sense, to attention updates and can *store* exponentially many patterns in the associative dimension with single-step retrieval (47). FFN layers in Transformers have also been interpreted as *key–value memories* (21). Our results complement this memory-centric view by isolating the *addressing* budget: how much key/query capacity is required to select the correct neighbors (edges) for arbitrary contexts. Together these views separate storage capacity from the cost of accurate retrieval/selection in the key–query channel.

## GRAPH TRANSFORMERS AND STRUCTURAL ENCODINGS

Expressivity of graph Transformers is shaped by structural encodings and higher-order tokenization. SEG-WL analyses show that structural features (e.g., SPIS encodings) set the attainable expressivity ceiling and can be matched by simple Transformer variants (72). Higher-order graph Transformers reach (or fall short of) $t$-WL power depending on whether explicit tuple indices and structural signals are provided (71). In the vision setting, (28) prove that ViTs can learn spatial structure under appropriate conditions, resonating with our assumptions that near-orthogonal embeddings and structural signals determine how efficiently edges can be packed and recovered; given such signals, our $D_K$-based bounds become tight predictors of success.

**Summary.** Across expressivity, connectivity, memorization, superposition, interpretability, memory equivalence, and graph structure, prior work identifies the ingredients that make attention powerful and the constraints that limit it. We contribute a capacity-centric bridge: a concrete relational task (RGR) in which the *total key dimension $D_K$* is the critical budget, with lower and upper bounds tight up to logarithmic factors, a principled multi-head advantage, and empirical thresholds that align with constructive algorithms.

# I   LLM USAGE STATEMENT

Throughout the preparation of this manuscript, we extensively utilized Large Language Models (LLMs) as assistive tools. Their application spanned several aspects of our workflow. For writing, LLMs were used to generate rough drafts from outlines and other notes, improve grammar and clarity, rephrase sentences, and refine the overall prose. In our software development, they served as coding assistants for generating boilerplate code, debugging, and refactoring our experimental scripts. Furthermore, LLMs were employed to accelerate our literature search by helping to identify relevant related work and suggesting key references. We also used them in a brainstorming capacity to ideate on potential experimental designs and ablation studies. The authors reviewed, edited, and take full responsibility for all content.

