# OpenReview forum: "On the Capacity of Self-Attention"
_ICLR.cc/2026/Conference — ICLR 2026 Conference Desk Rejected Submission_

### Official Review · Reviewer_TsWQ · 2025-10-17

**Soundness:** 3
**Presentation:** 1
**Contribution:** 2
**Rating:** 4
**Confidence:** 3

**Summary:**

The paper studies the capacity of self attention by considering the number of distinct relations a model can recover for a given budget (total query dimension). This is mathematically formalised by a ‘relational graph recognition’ (RGR) problem, where the task is to predict a node’s in-context neighbours (adjacent in the graph and with the same context label). For a given class of graphs, the authors ask: what is the minimum query dimension to get the RGR mapping for all graphs and all contexts? This is claimed to be a sensible abstraction for modelling how the modelling (c.f. memorisation, studied in previous work) capacity of attention varies with $d$.

**Strengths:**

1. I broadly buy that finding ways to quantify the modelling capacity of attention is an interesting research direction, and the theoretical results seem plausible.
2. The analysis of multi-head attention provides an interesting perspective on why introducing heads can help despite a fixed total head dimension.

**Weaknesses:**

1. The paper is clearly in violation of the formatting rules – the margins are _way_ smaller than they ought to be. My understanding is that this technically ought to lead to desk rejection. Of course, I defer to the AC on this point.
2. Whilst the theoretical results are plausible and Figs 1 and 2 verify that they hold in your idealised experimental setup, it would be helpful to see to what extent they hold/break down in more general settings – for more general graphs, using softmax, with unnormalised node embeddings. Even if the theory is intractable it would be good to get a sense of to what extent the empirical relationships still hold.
3. As a broader qualm, it’s not totally clear how this construction relates to observable properties of transformers. I very much appreciate rigorous, theoretical work, but think the arguments would be more convincing if the results could be related (even in a toyish setting) to actual observable behaviour in trained models on real data.

**Questions:**

See above.

1. To what extent will your empirical findings hold when you relax the assumptions? (A qualitative answer is fine, but imo you need some kind of answer)
2. Can you relate e.g. your scaling laws to any observable behaviour noted in transformers trained on real data -- either from your own experiments or from the literature?

---

> ### Author Response · Authors · 2025-11-20
>
> > **Formatting rules:**
>
> We apologize for the margin oversight. We have corrected the formatting in the revised PDF uploaded with this rebuttal.
>
> > **General settings (Softmax, unnormalized embeddings):**
>
> We have addressed Softmax in the Comments for all reviewers. Regarding unnormalized embeddings: Our theoretical results do not strictly require normalization. The "de-embedding" step in our construction can normalize inputs effectively. If the ratio of the largest to smallest norm is bounded, or if embeddings are drawn from a standard Gaussian $\mathcal{N}(0, I_{d_{model}})$, our proofs hold with only minor adjustments to constants and a global rescaling factor.
>
> > **Relation to observable properties / Scaling Laws:**
>
> We agree that bridging theory and practice is essential. While our construction is idealized, it models a very specific, observed phenomenon: **interference arising from superposition**.
>
> Our core thesis is that when the embedding dimension is compressed ($d_{model} \ll m$), relations cannot be stored orthogonally. This matches the 'superposition' observed in trained models (Elhage et al., 2022). In this regime, multiple heads act as a **noise-reduction device**, isolating specific subspaces to filter out interference from superposed features. This theoretical prediction maps to three observable properties:
>
> 1.  **Multi-Query Attention (MQA/GQA):** Empirical success with MQA (Shazeer, 2019) shows that models perform well even when sharing Values ($V$) and Keys ($K$) across heads, provided they have independent Queries ($Q$). Our theory explains *why*: the 'database' ($K$) is dense and superposed (noisy). The heads ($Q$) function as distinct filters acting on that same noisy data to extract different, clean relational subspaces.
> 2.  **The 'Superposition' Phenotype:** Mechanistic interpretability studies show that attention heads often perform specific tasks (e.g., 'Induction Heads' or 'Name Mover Heads') by attending to features that are polysemantic and superposed. Our construction formalizes the capacity cost of this operation: it quantifies exactly how many heads are required to 'read' a specific relationship out of a superposed basis without error.
> 3.  **Head Redundancy (Pruning):** The observation that many heads can be pruned (Voita et al., 2019) is often attributed to general over-parameterization. However, our capacity bounds suggest a more specific cause: if heads are noise-reduction devices, they are likely 'voting' or 'denoising' in parallel. Once the signal-to-noise ratio is high enough during training, some of these parallel denoisers become redundant, whereas a model storing unique content per head would be far less robust to pruning."

---

> > ### Author Response · Authors · 2025-11-24
> >
> > Another comment on the question regarding "Relation to observable properties in real networks":
> >
> > Our main result fits this description: we demonstrate that a large number of heads is required by an attention layer.  This is observed in practice, with many models utilizing 96 query heads.  It seems difficult to argue that there are 96 distinct, orthogonal semantic functions required at every layer of a large model.
> >
> > In more detail, we believe our results offer a necessary correction to the standard intuition regarding *why* multiple heads are effective in practice.
> > The conventional wisdom is that Multi-Head Attention (MHA) allows a model to “attend to information from different representation subspaces” (Vaswani et al., 2017)—implying that $h$ heads are needed to track $h$ distinct targets simultaneously. However, this intuition is contradicted by the widely observed phenomenon of head redundancy.
> >
> > Empirical studies have repeatedly shown that a large fraction of attention heads can be pruned at test time without performance degradation (Michel et al., 2019; Voita et al., 2019). If each head were attending to a unique, necessary target, such pruning would be catastrophic. Instead, this redundancy suggests that models frequently allocate multiple heads to attend to the same targe*. For instance, 'Induction Heads' often appear in groups, collectively attending to a single previous token to perform reliable copying (Olsson et al., 2022).
> >
> > Our results fit this observation perfectly. We demonstrate that when the basis is superposed (as it is in compressed embeddings), a large number of heads is required simply to filter interference. Thus, the high head counts observed in practice are likely driven not by the need to attend to 96 different things, but by the capacity required to reliably recover even a small number of relationships from a noisy basis.

---

### Official Review · Reviewer_d1fH · 2025-10-24

**Soundness:** 4
**Presentation:** 3
**Contribution:** 3
**Rating:** 6
**Confidence:** 3

**Summary:**

This paper investigates a fundamental question about self-attention mechanisms, given a computational budget, how many distinct relationships can a single attention layer reliable represent and recover? The authors formalize this through the Relational Graph Recognition framework, where attention must recover the edges of a directed graph among $m$ items $m'$ edges.

The key contributions are:

* Theoretical Bounds: The paper derives both information-theoretic lower bounds and constructive upper bounds for the minimum total key dimension D_K = h·d_k (number of heads × per-head key dimension) required to solve RGR. For most graph families, these bounds match up to logarithmic factors: D_K = Θ(m'log(m)/d_model).

* Multi-Head Advantage: The analysis reveals why multiple attention heads are beneficial even when each item only attends to a single target. When embeddings are compressed (d_model < m), relationships become superposed in overlapping subspaces. Multiple heads reduce interference by specializing on disjoint subsets of relationships.

* Explicit Constructions: The paper provides concrete algorithms showing how to construct attention weights that achieve the theoretical bounds for various embedding types (one-hot, Gaussian unit-norm, sparse binary).

* Empirical Validation: Experiments on permutation graphs show sharp phase transitions in performance as D_K increases, with the empirical threshold closely matching the theoretical prediction (R² = 0.944-0.992).

The work focuses on an idealized attention model that isolates the key-query computation while omitting components like softmax and the value pathway. This simplification enables precise mathematical analysis while preserving the essential computational constraints of attention.

**Strengths:**

* The paper provides both information-theoretic lower bounds and constructive upper bounds for self-attention capacity, with the bounds matching up to logarithmic factors in many cases. This is a significant theoretical contribution.

* The Relational Graph Recognition (RGR) framework is a clever abstraction that isolates the key-query mechanism while preserving the essential properties of self-attention. This allows for precise mathematical analysis.

* The experimental results closely match the theoretical predictions (R² = 0.944-0.992), providing strong evidence for the theory. The sharp phase transitions in performance as a function of D_K are particularly compelling.

* The paper provides a principled explanation for why multi-head attention is beneficial even when each item only attends to a single target - multiple heads reduce interference from superposition in compressed embeddings.

* The paper covers multiple embedding types (one-hot, Gaussian unit-norm, sparse binary) and extends from permutations to general graphs.

**Weaknesses:**

* The model omits several key components of real attention (softmax, value pathway). Furthermore, the OR of heads aggregation via max is not how standard multi-head attention works
* The degree-skew condition limits the tightness of bounds for certain graph families

**Questions:**

* How well do you expect these results to transfer to actual transformer architectures with all their components (softmax, values, layer norm, etc.)? Have you conducted any preliminary experiments with more realistic attention mechanisms?
* The analysis assumes fixed random embeddings. How would the capacit scaling change if embeddings could be learned to optimize for the graph structure? Could learned embeddings achieve better compression than random ones?
* What about the computational complexity of actually learning these attention patterns? Are there fundamental optimization challenges when operating near capacity?
* How can we inform ourselves based on the theory in the paper when designing real models containing attention? How should we pick the hyperparameters such as number of heads and head dimension etc.?

---

> ### Author Response · Authors · 2025-11-20
>
> > **Omission of Softmax/Value & Aggregation:**
>
> We have addressed this in the Comments for all reviewers. We now provide theory and experiments for Softmax and Value Channel extensions. Regarding aggregation: while standard MHA uses a sum, our "Max" aggregation was a theoretical proxy for non-linearity. Importantly, our new Softmax experiments use the standard **Sum** aggregation, and the results hold.
>
> > **Degree-skew condition:**
>
> This remains an interesting theoretical open problem. We suspect high degree-skew families are less relevant for the functional role of attention heads in practice.
>
> > **Transfer to actual transformer architectures:**
>
> Our new appendices show that the results transfer directly when adding Softmax and Value channels.
>
> > **Learned Embeddings:**
>
> With fixed random embeddings, we already match the information-theoretic lower bound within constant factors. Therefore, learned embeddings cannot improve the asymptotic capacity (as you cannot beat the lower bound). However, learned embeddings might allow the model to find these efficient packings more easily.
>
> > **Computational Complexity:**
>
> The constructive algorithms we describe have computational complexity $O(m d_{model} \log m)$, which is significantly cheaper than training via gradient descent. We did not encounter fundamental optimization challenges near capacity, though convergence takes longer in highly compressed regimes ($m \gg d_{model}$).
>
> > **Impact of theory to Hyperparameters and other design choices:**
>
> Our theory and its validation in our toy models is prescriptive here:
> 1.  **Number of heads:** Should scale with the compression factor $m' / d_{model}$.
> 2.  **Head dimension:** Should be small, roughly logarithmic in the number of edges ($\log m'$, or $\log^2 m'$ with the addition of softmax and scaling.
> 3.  **Non-linearity:** This is essential for noise control (whether Softmax or Max over heads).

---

### Official Review · Reviewer_B9jt · 2025-10-28

**Soundness:** 2
**Presentation:** 3
**Contribution:** 2
**Rating:** 4
**Confidence:** 3

**Summary:**

This paper introduces a formal framework, Relational Graph Recognition (RGR), to analyze the "capacity" of a single self-attention layer. Capacity is defined as the number of distinct relations ($m'$) that a layer can reliably recover. The paper's main theoretical contribution is a capacity scaling law, $D_K = \Theta(m' \log m' / d_{model})$, where $D_K = h \cdot d_k$ is the total key dimension (number of heads times per-head key dimension). This analysis provides a new rationale for multi-head attention: when embeddings are compressed ($m > d_{model}$), distributing the $D_K$ budget across multiple heads mitigates interference and increases capacity. The paper provides an information-theoretic lower bound and a constructive upper bound to establish this scaling law, and validates it with experiments on synthetic data.

**Strengths:**

1.  **Novel Formalization:** The paper provides a novel and interesting perspective on attention capacity. The RGR framework and the $D_K$ (total key dimension) metric are a clear and useful way to formalize the problem of relational storage in the QK-circuit.
2.  **Rigorous Theoretical Analysis:** Within its stated assumptions, the theoretical analysis is comprehensive and sound. The authors provide both a lower bound (necessity) and an explicit construction (sufficiency) to derive their scaling law, which is a complete line of theoretical argument.
3.  **New Rationale for Multi-Head Attention:** The explanation that multiple heads help mitigate interference from *embedding compression* (rather than just attending to different things) is an elegant and intuitive contribution to the understanding of MHA.

**Weaknesses:**

1.  **Over-simplification (Omission of Softmax):** The most significant weakness is the decision to model attention using a global threshold on max-pooled scores, thereby completely omitting the softmax function. This simplification is not minor; it fundamentally changes the mechanism's behavior. Softmax introduces dynamic, context-dependent competition among tokens. The paper's static, non-competitive model is a very different mechanism.
2.  **Lack of Relevance to LLMs:** Because of this simplification, the derived theoretical bounds are almost meaningless for practical LLM research. A large body of work on "linear attention" (softmax-free models) has repeatedly shown that while these models are efficient, they suffer from severe performance degradation precisely because they lack the competitive, non-linear dynamics of softmax. This paper's theory applies to a class of models that are known to be less powerful in practice.
3.  **Purely "Simulation Validation":** The experimental setup does not test the theory's applicability; it merely validates the theory *within its own idealized assumptions*. The experiments are run on synthetic permutation graphs using an idealized model that perfectly matches the theoretical construction (no softmax, no OV circuit). The successful validation is therefore almost a foregone conclusion and feels more like a "simulation" of the theory than an empirical validation of its relevance.
4.  **No Real-Data Experiments:** The paper lacks *any* experiments on real-world data, even a simple task. This is a critical omission for a deep learning conference. Without evidence that these findings hold (or even correlate) on a real dataset, the paper reads more like a mathematical analysis report on a simplified algorithm than a deep learning paper with practical implications. It is crucial to demonstrate that this theoretical capacity is the *relevant* capacity for tasks we care about.

**Questions:**

1.  The core critique lies in the removal of softmax. How do the authors believe their $D_K$ bounds would change in the presence of softmax-induced competition? Would the "interference" from embedding compression still be the dominant factor, or would the context-dependent competition render these bounds inapplicable?
2.  Given the well-documented performance drop of linear (softmax-free) attention models, it suggests their "capacity" is insufficient for complex tasks like language modeling. How does the paper's definition of capacity (recovering graph edges) relate to the functional capacity required for such tasks?
3.  Why did the authors not attempt even a preliminary experiment on a real-world dataset (e.g., WikiText) to show that their $D_K$ metric correlates with downstream performance or that the multi-head advantage manifests as predicted? This would significantly strengthen the paper's claims of relevance.

---

> ### Author Response · Authors · 2025-11-20
>
> > **Omission of Softmax & Relevance to LLMs:**
>
> We appreciate this critique as it pushed us to rigorously verify our claims. As detailed in the Comments for all reviewers, we have extended the theory and experiments to include Softmax. The results are robust: interference from embedding compression remains the dominant factor, and the multi-head advantage persists.
>
> > **Connection to Linear Attention:**
>
> This is a vital distinction. Our model is **not** linear attention because of the *max-over-heads* non-linearity. In fact, our theory offers a new justification for *why* linear attention (which lacks this non-linearity) fails to scale: it cannot localize noise. By showing that *some* non-linearity is required to unlock the capacity of compressed embeddings, our work supports the necessity of Softmax (or some other non-linearity) in LLMs.
>
> > **"Simulation Validation" vs. Real Data:**
>
> While we agree real-world data is the ultimate test, the experiments here are not mere simulations of our algorithms. We train the models via gradient descent. The fact that GD *finds* these optimal solutions (matching our constructive bounds) is a non-trivial validation of the optimization landscape, not just the existence of a solution.
>
> > **Lack of Real-Data Experiments:**
>
> We agree that verifying these laws on real LLMs is the next step. However, mechanistic interpretability often requires isolating variables to establish baselines. Real-world data introduces confounders (feature learning, superposition, layer norm interactions) that make measuring raw capacity difficult. Our work establishes the "theoretical speed limits" against which real models can be measured.

---

> > ### Comment · Reviewer_B9jt · 2025-11-27
> >
> > Thank you for your extension of softmax, which makes this work more complete. Overall, I think this is a good theoretical work, but it lacks the necessary (can be small-scale) experiments on real data to validate the theory. I maintain my rating and suggest that the authors supplement real data experiments to make it a strong submission for the next time.

---

### Official Review · Reviewer_EVn9 · 2025-10-28

**Soundness:** 3
**Presentation:** 2
**Contribution:** 3
**Rating:** 6
**Confidence:** 3

**Summary:**

This paper presents a formal theoretical analysis of the *capacity* of self-attention layers—the number of distinct token-to-token relations that a single attention layer can reliably represent, given a fixed parameter budget. To formalize this, the authors introduce the **Relational Graph Recognition (RGR)** framework, where the key–query channel is viewed as encoding a directed graph ( G = (V, E) ) among ( m ) items. The capacity problem is then cast as determining the minimal total key dimension required to recover all ( m' ) relations. The paper provides the first quantitative scaling law linking relational capacity, embedding dimension, and number of heads in self-attention.

**Strengths:**

* The RGR formulation is novel and mathematically elegant, bridging mechanistic interpretability and information-theoretic perspectives. Unlike prior “memorization capacity” or “expressivity” analyses, RGR isolates *where* attention attends (the QK channel) rather than *what* it writes (the OV path), leading to fundamentally new capacity laws. The theoretical connection between *multi-head specialization* and *noise reduction under compression* is an insightful reinterpretation of the design rationale for multi-head attention.


* The paper provides both **lower bounds** (information-theoretic) and **constructive upper bounds** (algorithmic proofs) that nearly match, supported by precise asymptotics and concentration arguments.
* Empirical experiments are well-controlled and directly aligned with the theoretical model (frozen embeddings, synthetic graphs, fixed thresholds), validating the scaling laws quantitatively (( R^2 > 0.99 )).

**Weaknesses:**

* There is no equation number for all equations.

* Experiments are restricted to synthetic permutation graphs (out-degree = 1) and omit evaluation on multi-edge graphs (( \Delta > 1 )). Testing richer relational structures or real transformer layers would strengthen the empirical credibility.

* The omission of softmax normalization and the value pathway isolates QK capacity but makes the model somewhat detached from actual transformer dynamics. A discussion of whether the same scaling persists under full attention (with softmax) would be valuable. Or experiments validate the conclusion with softmax.

**Questions:**

* How is the directed graph built? Full attention allows tokens to attend to all other tokens so it should be a complete graph which is very dense. But this could add a gap of logm according to the paper. So how will this affect the application of the theory in real-world tasks?

---

> ### Author Response · Authors · 2025-11-20
>
> **Response to Reviewer EVn9**
>
> > **Equation numbers:**
>
> We have updated the revision to ensure all equations are numbered for easier referencing.
>
> > **Testing richer relational structures / Effect of dense graphs:**
>
> As detailed in the Comments for all reviewers, we have performed additional experiments on denser ($r$-regular) graphs. These strongly validate the edge-based capacity view: both capacity and optimal head count are governed by edge count ($m'$), not vertex count ($m$).
>
> > **Omission of Softmax:**
>
> We have addressed this in the new appendices. We show theoretically and empirically that the scaling laws and multi-head advantage persist with Softmax (picking up only a logical $\log m$ factor due to scaling).
>
> > **Graph sparsity in real-world tasks:**
>
> You ask about the "complete graph" of attention. While attention *computes* a score for all pairs, the "relational graph" we model represents the subset of tokens that *should* be attended to (e.g., a noun attending to its modifier). In practice, if a head attends to everything uniformly, the signal is washed out. Therefore, the "ground truth" graph for any meaningful attention head is likely sparse, aligning with our model, and likely to be in the regime where our upper and lower bounds asymptotically match.

---

> > ### Comment · Reviewer_EVn9 · 2025-11-21
> >
> > Thank you for your responses. My concerns have been addressed and I will keep my score.

---

### Author Response · Authors · 2025-11-20
**Comments for all reviewers focused on model extensions**

First, thank you for the careful, constructive, and genuinely insightful feedback. Regardless of the final decision, your comments helped us clarify the scope of our results, tighten the presentation, and make the claims easier to connect to standard transformer practice.

A common theme across reviews (EVn9, B9jt, d1fH, TsWQ) was whether our conclusions persist once we include aspects of the standard attention mechanism that were omitted in the simplified model—specifically **Softmax**, the **Value Channel**, and **Denser Graphs**.

We agree this is crucial. We have uploaded a revision with three new appendices: "Extension to Softmax," "Incorporating a Value Channel," and "Experiments with Model Extensions." We also report new experiments on denser graphs. These results demonstrate that our core findings are robust and are not artifacts of the simplified model.

**1. What changes (and what does not) when we add Softmax?**
Our central conclusions are robust to the inclusion of Softmax.

* **Lower Bounds:** Unchanged. The softmax variant preserves the Lipschitzness of the decision function with respect to parameters, which is at the heart of our information-theoretic lower bound.   Intuitively, softmax does not increase how many labelings the parameters can encode; it only changes how a margin is converted into a probability.
* **Constructive Upper Bound:** We adapt Construction II to the softmax model. We show that nearly the same scaling law applies: $D_K = O(\frac{m}{d_{model}}(\log m)^2)$. The additional $\log m$ factor is a necessary consequence of the standard scaling used inside the softmax logits.
* **Multi-head Advantage:** This persists. In the compressed regime ($m \gg d_{model}$), the capacity-based rationale for many small heads survives.
* **Experiments:** We repeated the controlled experiments under the softmax model (using standard softmax and head-sum aggregation). The F1–$D_K$ curves again show a sharp phase transition and a clear multi-head advantage, consistent with the theoretical prediction.

**2. Comparison to Linear Attention (Response to B9jt)**
Reviewer B9jt highlighted a connection to "linear attention." It is critical to clarify that our core model is **not** linear attention.
* **The Difference:** Linear attention typically implies linearity over the context and a *sum* over heads. Our model uses a *max-over-heads* aggregation. This non-linearity is crucial: it allows heads to specialize and localize interference.
* **Why it matters:** Our analysis shows that a non-linearity is required to preserve noise isolation. When we study (both theoretically and empirically) a strictly linear variant of our model (sum-over-heads), the multi-head advantage disappears, and performance degrades significantly.  Specifically, the capacity for multiple heads becomes the same as for a single head.
* **Implication:** Our theory provides a new, capacity-based explanation for *why* linear attention often underperforms in practice: without a non-linearity (be it Softmax or Max-over-heads) to gate interference, the mechanism loses the noise-reduction benefits of multi-head architectures.

**3. What changes when we add a Value Channel?**
We added a new appendix ("Incorporating a Value Channel") where we augment the softmax version of the model with a standard value path ($W_V$) and linear output projection ($W_O$).
* **Result:** The key–query budget remains unchanged: $D_K=\Theta(\frac{m}{d_{model}}(\log m)^2)$, and the multi-head advantage persists.
* **Experiments:** We also repeated the controlled experiments with the standard value path added to the core model.  We once again see that the multi-head advantage persists.
* **Why:** The value channel affects constants and retrieval rates but not the central conclusion: the bottleneck is QK separation, and combining a nonlinearity (softmax or max over heads) with many small heads remains the right capacity principle.

**4. Experiments with Denser Graphs**
We generalized our data generation to directed $r$-regular graphs, scaling the degree $r$ from 1 to 32.
* **Edge-based Capacity:**  We found that capacity is governed by the number of edges ($m'$), not vertices ($m$). When normalized by $m'$, transition thresholds for different densities collapse onto a single linear trend.  This confirms our theoretical results that the "where-to-attend" budget scales with the number of encoded relationships.
* **Head Count:** The optimal number of heads grows linearly with density ($r$), confirming our prediction that head count should scale with $m'/d_{model}$.  The multi-head advantage is even more pronounced in denser graphs.

---

### Author Response · Authors · 2025-12-03
**Thank You / Summary for New Area Chair**

Dear Area Chair,

Thank you for overseeing this process, especially in light of the additional complexity resulting from this year's process.

Very briefly, our paper studies the capacity of the self-attention mechanism to represent relationships, and derives nearly matching upper and lower bounds on the required dimension of the key and query matrices of attention. This leads to a capacity-based explanation of why many small heads are beneficial, demonstrating that the standard story of "multiple heads are required to pay attention to multiple things simultaneously" is incomplete.  This analysis is reinforced through controlled experiments.  All four reviewers viewed the direction as interesting and the theoretical results as plausible or strong; the main reservations were about **model realism and empirical relevance** rather than correctness.

In response to these reviews, we uploaded a revised version with **new theory and experiments** directly targeting those concerns. Below we summarize what changed and how it addresses the reviewers’ main points.

---

### A. Adding Softmax: theory and experiments

**Concern (B9jt, EVn9, d1fH, TsWQ):**
The core criticism was that omitting softmax makes the model too idealized and potentially irrelevant to real transformers. Reviewer B9jt in particular argued that this omission makes our bounds “almost meaningless” for LLMs.

We added an **“Extension to Softmax”** appendix showing that the information-theoretic lower bound is unchanged in the softmax setting, and adapt our constructive upper bound, obtaining essentially the same scaling law (up to an additional $\log m$ factor from softmax scaling). We also reran the synthetic experiments with softmax and found the same **sharp phase transition** in F1 as a function of $D_K$, and found that the **multi-head advantage persists**: many small heads outperform a single large head, in line with the theory. Thus, our main conclusions are robust to including softmax.

**Effect on concerns:**
This directly addresses the main reason Reviewer B9jt rated the paper just below the acceptance threshold and also responds to EVn9/d1fH/TsWQ’s questions about robustness of the conclusions under more realistic attention mechanisms.

---

### B. Incorporating the Value Channel

**Concern (B9jt, d1fH, TsWQ):**
The original model neglected the value path and output projection; reviewers asked whether this would change the conclusions.

We added a new appendix **“Incorporating a Value Channel”**. We there extend the softmax model with a standard value matrix, and show that while these components affect constants and retrieval dynamics, the **bottleneck remains the QK channel**, the main scaling law still holds and there is still a clear multi-head advantage for compressed embeddings. These analytical results are reinforced by further synthetic experiments in this richer setting.

**Effect on concerns:**
This shows that our main claims are not an artifact of omitting the value path; they are specifically about **where capacity is bottlenecked** (QK separation), which remains true in a more standard transformer-style attention block.

---

### C. Denser Graphs and richer relational structures

**Concern (EVn9, TsWQ, d1fH):**
Experiments were originally restricted to degree 1 graphs. Reviewers asked whether the theory and empirical findings extend to **higher degree graphs** and richer relational structures.

The new draft includes experiments beyond permutation graphs to **directed $r$-regular graphs** with degrees up to 32. Capacity and optimal head count scale with the **number of edges $m'$** rather than just vertices $m$; when we normalize by $m'$, transition thresholds for different densities collapse onto a single linear trend, as predicted by our theory. The multi-head advantage becomes even more pronounced in denser graphs.

**Effect on concerns:**
This directly addresses EVn9’s request for $\Delta > 1$ settings and TsWQ’s concern about more general relational structures, and strengthens the practical relevance of our capacity picture.

---

> ### Author Response · Authors · 2025-12-03
> **Thank You / Summary for New Area Chair (part 2)**
>
> ### D. Relation to linear attention
>
> **Concern (B9jt):**
> Reviewer B9jt described our model as **linear attention**, citing the empirical weaknesses of such models and arguing that our analysis may thus have limited relevance for LLMs.
>
> We clarify that our core model is **not linear attention**. Linear attention models are linear in the context and effectively a **sum over heads**, whereas our construction uses a **max-over-heads nonlinearity**, which is crucial for **head specialization and interference control**. Our model can analyze a **strictly linear variant** (sum-over-heads), and both theory and experiments show that in this linear setup, the **multi-head advantage vanishes** — capacity with many heads collapses to that of a single head. This offers a new, capacity-based explanation for **why linear attention often underperforms**: without a nonlinearity (softmax over the context or max over heads), the mechanism cannot localize noise, and the benefits of multiple heads disappear.
>
> **Effect on concerns:**
> Together with the softmax/value extensions, this substantially strengthens the **relevance of our results to modern LLM architectures** and directly addresses Reviewer B9jt’s core objections.
>
> ---
>
> ### E. Experiments on large models
>
> All four reviewers also noted the absence of experiments on large, real-world transformer models. We fully agree that this is an important direction, but we also believe it is beyond the intended scope of this paper. Our goal here is to provide *clean capacity limits* in an idealized setting, so that we can explain and organize phenomena that have already been repeatedly observed in practice. Our results provide analytical justification for **why** multiple small heads are needed, which has been found to be crucial in all large language models. Our model's predictions are also consistent with other observed phenomena, such as the shortcomings of linear attention, head redundancy, and Multi-Query Attention.
>
> Thus, we see this work as sitting squarely in the **theoretical / mechanistic** part of the ICLR scope. We believe there is a genuine need for work that cleanly characterizes fundamental limits and then connects them to the behaviour of existing models, without conflating those questions with the many confounders of end-to-end training. In that sense, we hope this submission can be viewed less as “missing a real-networks experiment checkbox” and more as **providing the theoretical lens** through which such experiments (many of which already exist) can be interpreted.
>
> ---
>
> ### F. Presentation and formatting
>
> **Concern (TsWQ, EVn9):**
> Missing equation numbers, violations of formatting/margin rules.
>
> The revised PDF now numbers all equations for easier cross-referencing, and corrects the margins and formatting to conform fully to ICLR guidelines.
>
> **Effect on concerns:**
> These were important but purely presentational; they are now resolved.
>
> ---
>
> We respectfully submit that the revised version now offers:
>
> 1. A **novel and technically strong** capacity theory for self-attention;
> 2. A **robust explanation** for multi-head attention that is stable under more realistic modeling choices; and
> 3. A **theoretically grounded account** of several empirical transformer phenomena (head counts, redundancy, superposition, MQA) that are already widely observed.
>
> Given that the theoretical core was already viewed positively by all reviewers, and that the revision directly targets the main realism and empirical-relevance concerns behind the two 4-scores, we hope that the paper now clearly meets the acceptance bar.
>
> Thank you again for your time and for considering this summary in your decision.

---

### Note · Program_Chairs · 2026-01-17
**Submission Desk Rejected by Program Chairs**

The following references in this submission do not refer to real documents and/or have major errors in bibliographic information:

 Gal Vardi, Gilad Yehudai, and Ohad Shamir. Memorization thresholds in deep neural networks. arXiv preprint arXiv:2002.10211, 2020.